# Discovering Non-monotonic Autoregressive Orderings with Variational Inference

**Xuanlin Li**,* **Brandon Trabucco**,* **Dong Huk Park,  Michael Luo**
University of California, Berkeley
{xuanlinli17, btrabucco, dong.huk.park, michael.luo}@berkeley.edu

**Sheng Shen,  Trevor Darrell,  Yang Gao**
University of California, Berkeley; Tsinghua University
{sheng.s, trevordarrell}@berkeley.edu, gy20073@gmail.com

## Abstract

The predominant approach for language modeling is to encode a sequence of tokens from left to right, but this eliminates a source of information: the order by which the sequence was naturally generated. One strategy to recover this information is to decode both the *content* and *ordering* of tokens. Some prior work supervises content and ordering with hand-designed loss functions to encourage specific orders or bootstraps from a predefined ordering. These approaches require domain-specific insight. Other prior work searches over valid insertion operations that lead to ground truth sequences during training, which has high time complexity and cannot be efficiently parallelized. We address these limitations with an unsupervised learner that can be trained in a fully-parallelizable manner to discover high-quality autoregressive orders in a data driven way without a domain-specific prior. The learner is a neural network that performs variational inference with the autoregressive ordering as a latent variable. Since the corresponding variational lower bound is not differentiable, we develop a practical algorithm for end-to-end optimization using policy gradients. Strong empirical results with our solution on sequence modeling tasks suggest that our algorithm is capable of discovering various autoregressive orders for different sequences that are competitive with or even better than fixed orders.

## 1 Introduction

Autoregressive models have a rich history. Early papers that studied autoregressive models, such as (Uria et al., 2016) and (Germain et al., 2015), showed an interest in designing algorithms that did not require a gold-standard autoregressive order to be known upfront by researchers. However, these papers were overshadowed by developments in natural language processing that demonstrated the power of the left-to-right autoregressive order (Cho et al., 2014; Sutskever et al., 2014a). Since then, the left-to-right autoregressive order has been essential for application domains such as image captioning (Vinyals et al., 2015b; Xu et al., 2015), machine translation (Luong et al., 2015; Bahdanau et al., 2015) and distant fields like image synthesis (van den Oord et al., 2016). However, interest in non left-to-right autoregressive orders is resurfacing (Welleck et al., 2019b; Stern et al., 2019), and evidence (Vinyals et al., 2016; Gū et al., 2018; Alvarez-Melis & Jaakkola, 2017) suggests adaptive orders may produce more accurate autoregressive models. These positive results make designing algorithms that can leverage adaptive orders an important research domain.

Inferring autoregressive orderings in a data-driven manner is challenging. Modern benchmarks for machine translation (Stahlberg, 2019) and other tasks (Oda et al., 2015) are not labelled with gold-standard orders, and left-to-right seems to be the default. This could be explained if domain-independent methodology for identifying *high-quality* orders is an open question. Certain approaches (Stern et al., 2019; Welleck et al., 2019b; Ruis et al., 2020) use hand-designed loss functions to promote a *genre* of orders—such as balanced binary trees. These loss functions incorporate

---

*Authors contributed equally.

certain domain-assumptions: for example, they assume the balanced binary tree order will not disrupt learning. Learning disruption is an important consideration, because prior work shows that poor orders may prohibitively slow learning (Chen et al., 2018). Future approaches to inferring autoregressive orders should withhold domain knowledge, to promote their generalization.

To our best knowledge, we propose the first domain-independent unsupervised learner that discovers high-quality autoregressive orders through fully-parallelizable end-to-end training without domain-specific tuning. We provide three main contributions that stabilize this learner. First, we propose an encoder architecture that conditions on training examples to output autoregressive orders using techniques in combinatorical optimization. Second, we propose *Variational Order Inference* that learns an approximate posterior over autoregressive orders. Finally, we develop a practical algorithm for solving the resulting non-differentiable ELBO end-to-end with policy gradients.

Empirical results with our solution on image captioning, code generation, text summarization, and machine translation tasks suggest that with similar hyperparameters, our algorithm is capable of recovering autoregressive orders that are even better than fixed orders. Case studies suggest that our learned orders depend adaptively on content, and resemble a type of *best-first* generation order, which first decodes focal objects and names. Our experimental framework is available at this link.

## 2  RELATED WORKS

**Autoregressive Models** Autoregressive models decompose the generation of a high dimensional probability distribution by generating one dimension at a time, with a predefined order. Combined with high capacity neural networks, this approach to modeling complex distributions has been very successful (Sutskever et al., 2011; Mikolov et al., 2012). Recent works have achieved great improvements with autoregressive models in many applications, including language modeling (Radford et al., 2018; 2019; Brown et al., 2020), machine translation (Sutskever et al., 2014b) and image captioning (Karpathy & Fei-Fei, 2015). Most previous works on autoregressive models use a fixed ordering pre-defined by the designer with left-to-right emerging as the primary choice. In contrast, our method is capable of learning arbitrary orderings conditioned on data and is more flexible.

**Non-Monotonic Autoregressive Orderings** Ford et al. (2018b) shows that a sub-optimal ordering can severely limit the viability of a language model and propose to first generate a partially filled sentence template and then fill in missing tokens. Previous works have also studied bidirectional decoding (Sun et al., 2017; Zhou et al., 2019; Mehri & Sigal, 2018) and syntax trees based decoding (Yamada & Knight, 2001; Charniak et al., 2003; Dyer et al., 2016; Aharoni & Goldberg, 2017; Wang et al., 2018) in the natural language setting. However, all of the works mentioned above do not learn the orderings and instead opt to use heuristics to define them. Chan et al. (2019) performs language modeling according to a known prior, such as balanced binary tree, and does not allow arbitrary sequence generation orders. Welleck et al. (2019a) proposes to use a tree-based recursive generation method to learn arbitrary generation orders. However, their performance lags behind that of left-to-right. Gu et al. (2019a) proposes Transformer-InDIGO to allow non-monotonic sequence generation by first pretraining with pre-defined orderings, such as left-to-right, then fine-tuning use Searched Adaptive Order (SAO) to find alternative orderings. They report that without pretraining, the learned orders degenerate. In addition, they perform beam search when decoding each token during training, which cannot be efficiently parallelized at the sequence length dimension. Emelianenko et al. (2019) proposes an alternative to SAO, but suffers from similar poor time complexity. In contrast, our method learns high-quality autoregressive orderings directly from data under fully-parallelizable end-to-end training.

**Variational Methods** Our method optimizes the evidence lower bound, or ELBO in short. ELBO is a quantity that is widely used as an optimization proxy in the machine learning literature, where the exact quantity is hard to compute or optimize. Variational methods have achieved great success in machine learning, such as VAE (Kingma & Welling, 2013) and $\beta$-VAE (Higgins et al., 2017).

**Combinatorial Optimization** Recent works have studied gradient-based optimization in the combinatorial space of permutations (Mena et al., 2018; Grover et al., 2019; Linderman et al., 2018). These works have been applied in tasks such as number sorting, jigsaw puzzle solving, and neural signal identification in worms. To our best knowledge, we are the first to build on these techniques to automatically discover autoregressive orderings in vision and language datasets.

## 3  PRELIMINARIES

The goal of autoregressive sequence modelling is to model an ordered sequence of target values $\mathbf{y} = (y_1, y_2 \ldots, y_n) : y_i \in \mathbb{R}$, possibly conditioned on an ordered sequence of source values $\mathbf{x} = (x_1, x_2 \ldots, x_m) : x_i \in \mathbb{R}$, where $(\mathbf{x}, \mathbf{y})$ is sampled from the dataset $\mathcal{D}$.

Inspired by Vinyals et al. (2015a) and Gu et al. (2019a), we formulate the generation process of $\mathbf{y}$ as a $2n$ step process, where at time step $2t - 1$ we generate a value, and at timestep $2t$ we select a not-yet-chosen position in $\{1, 2, \cdots, n\}$ to insert the value. Thus, we introduce the latent sequence variable $\mathbf{z} = (z_1, z_2 \ldots, z_n) : \mathbf{z} \in S_n$, where $S_n$ is the set of one-dimensional permutations of $\{1, 2, \cdots, n\}$, and $z_t$ is defined as the absolute position of the value generated at time step $2t - 1$ in the naturally ordered $\mathbf{y}$. Then $p(\mathbf{y}, \mathbf{z}|\mathbf{x})$ denotes the probability of generating $\mathbf{y}$ in the ordering of $\mathbf{z}$ given the source sequence $\mathbf{x}$. We can thus factorize $p(\mathbf{y}, \mathbf{z}|\mathbf{x})$ using the chain rule:

$$p(\mathbf{y}, \mathbf{z}|\mathbf{x}) = p(y_{z_1}|\mathbf{x})p(z_1|y_{z_1}, \mathbf{x}) \prod_{i=2}^{n} p(y_{z_i}|z_{<i}, y_{z_{<i}}, \mathbf{x})p(z_i|z_{<i}, y_{z_{<=i}}, \mathbf{x}) \tag{1}$$

For example, $p(y_1, y_2, z_1 = 2, z_2 = 1|\mathbf{x}) = p(y_2|\mathbf{x})p(z_1|y_2, \mathbf{x})p(y_1|z_1, y_2, \mathbf{x})p(z_2|y_1, z_1, y_2, \mathbf{x})$ is defined as the probability of generating $y_2$ in the first step, then inserting $y_2$ into absolute position 2, then generating $y_1$, and finally inserting $y_1$ into absolute position 1.

Note that in practice, the length of $\mathbf{y}$ is usually varied. Therefore, we do not first create a fixed-length sequence of blanks and then replace the blanks with actual values. Instead, we dynamically insert a new value at a position relative to the previous values. One common approach to predict such relative position is Pointer Network (Vinyals et al., 2015a). In other words, at timestep $t$, we insert the value at position $r_t$ relative to the previous generated values. Here, for any $\mathbf{z} \in S_n$, $\mathbf{r} = (r_1, r_2, \ldots, r_n)$ is constructed such that there is a bijection between $S_n$ and the set of all constructed $\mathbf{r}$. Due to such bijection, we can use $\mathbf{z}$ and $\mathbf{r}$ interchangeably. We will use $\mathbf{z}$ throughout the paper.

## 4  VARIATIONAL ORDER INFERENCE (VOI)

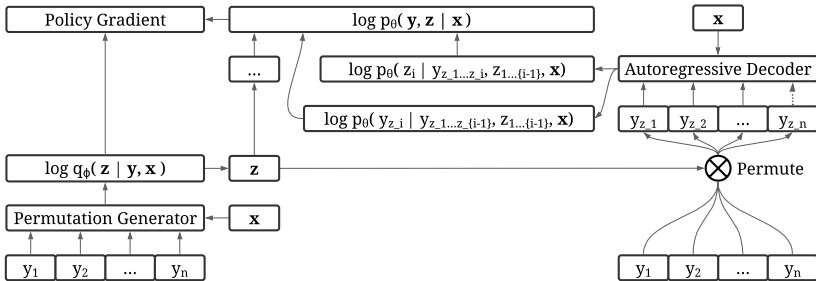

Figure 1: Computational Graph for Variational Order Inference

Starting from just the original data $\mathbf{y}$ in natural order, we can use variational inference to create an objective (2) that allows us to recover latent order $\mathbf{z}$, parametrized by two neural networks $\theta$ and $\phi$. The encoder network $\phi$ samples autoregressive orders given the ground truth data, which the decoder network $\theta$ uses to recover $\mathbf{y}$. More specifically, $\phi$ is a non-autoregressive network (permutation generator in Fig. 1) that takes in the source sequence $\mathbf{x}$ and the entire ground truth target sequence $\mathbf{y}$ and outputs latent order $\mathbf{z}$ in a single forward pass. $\theta$ is an autoregressive network (autoregressive decoder in Fig. 1) that takes in $\mathbf{x}$ and predicts both the target sequence $\mathbf{y}$ and the ordering $\mathbf{z}$ through the factorization in Equation (1). We name this process *Variational Order Inference* (VOI).

$$\mathbb{E}_{(\mathbf{x},\mathbf{y})\sim\mathcal{D}}\left[\log p_\theta(\mathbf{y}|\mathbf{x})\right] = \mathbb{E}_{(\mathbf{x},\mathbf{y})\sim\mathcal{D}}\left[\log \mathbb{E}_{\mathbf{z}\sim q_\phi(\mathbf{z}|\mathbf{y},\mathbf{x})}\left[\frac{p_\theta(\mathbf{y},\mathbf{z}|\mathbf{x})}{q_\phi(\mathbf{z}|\mathbf{y},\mathbf{x})}\right]\right]$$
$$\geq \mathbb{E}_{(\mathbf{x},\mathbf{y})\sim\mathcal{D}}\left[\mathbb{E}_{\mathbf{z}\sim q_\phi(\mathbf{z}|\mathbf{y},\mathbf{x})}\left[\log p_\theta(\mathbf{y},\mathbf{z}|\mathbf{x})\right] + \mathcal{H}_{q_\phi}(\cdot|\mathbf{y},\mathbf{x})\right] \tag{2}$$

Here, $\mathcal{H}_{q_\phi}$ is the entropy term. During training, we train $\phi$ and $\theta$ jointly to maximize the ELBO in (2). During testing, we only keep the decoder $\theta$.

To optimize the decoder network $\theta$ in (2), for each $\mathbf{y}$, we first sample $K$ latents $\mathbf{z}_1, \mathbf{z}_2, \ldots, \mathbf{z}_K$ from $q_\phi(\cdot|\mathbf{y}, \mathbf{x})$. We then update $\theta$ using the Monte-Carlo gradient estimate $\mathbb{E}_{\mathbf{y} \sim \mathcal{D}} \left[ \frac{1}{K} \sum_{i=1}^{K} \nabla_\theta \log p_\theta(\mathbf{y}, \mathbf{z}_i|\mathbf{x}) \right]$.

---

**Algorithm 1** Variational Order Inference

1: **Given:** encoder network $\phi$ with learning rate $\alpha_\phi$, decoder network $\theta$ with learning rate $\alpha_\theta$, entropy coefficient $\beta$, batch of training data $(\mathbf{X}, \mathbf{Y}) = \{(\mathbf{x}_b, \mathbf{y}_b)\}_{b=1}^{N}$ sampled from dataset $\mathcal{D}$
2: Set gradient accumulators $g_\phi = 0$, $g_\theta = 0$
3: **for** $(\mathbf{x}, \mathbf{y}) \in (\mathbf{X}, \mathbf{Y})$ **do**          $\triangleright$ In practice, this is done through parallel tensor operations
4:      $X = \phi(\mathbf{y}, \mathbf{x})$
5:      Sample $K$ doubly stochastic matrices $B_1, B_2, \ldots, B_K \in \mathcal{B}_{n \times n}$ from $\mathcal{G.S.}(X, \tau)$
6:      Obtain $P_1, P_2, \ldots, P_K \in \mathcal{P}_{n \times n}$ from $B_1, B_2, \ldots, B_K$ using Hungarian Algorithm
7:      Obtain latents $\mathbf{z}_1, \mathbf{z}_2, \ldots, \mathbf{z}_K = f_{\text{len}(\mathbf{y})}^{-1}(P_1), f_{\text{len}(\mathbf{y})}^{-1}(P_2), \ldots, f_{\text{len}(\mathbf{y})}^{-1}(P_K)$
8:      $g_\theta = g_\theta + \frac{1}{N \cdot K} \sum_{i=1}^{K} \nabla_\theta \log p_\theta(\mathbf{y}, \mathbf{z}_i|\mathbf{x})$
9:      Calculate $\log q_\phi(\mathbf{z}_i|\mathbf{y}, \mathbf{x}) = \langle X, P_i \rangle_F - \log(\text{perm}(\exp(X)))$
          $\approx \langle X, P_i \rangle_F - \log(\text{perm}_B(\exp(X)))$
10:      Calculate $b(\mathbf{y}, \mathbf{x}) = \frac{1}{K} \sum_{i=1}^{K} \log p_\theta(\mathbf{y}, \mathbf{z}_i|\mathbf{x})$
11:      $g_\phi = g_\phi + \frac{1}{N \cdot K} \sum_{i=1}^{K} \nabla_\phi \log q_\phi(\mathbf{z}_i|\mathbf{y}, \mathbf{x})(\log p_\theta(\mathbf{y}, \mathbf{z}_i|\mathbf{x}) - b(\mathbf{y}, \mathbf{x})) + \beta \cdot \nabla_\phi \mathcal{H}_{q_\phi}(\cdot|\mathbf{y}, \mathbf{x})$
12: **end for**
13: $\phi = \phi + \alpha_\phi \cdot g_\phi$
14: $\theta = \theta + \alpha_\theta \cdot g_\theta$

---

Optimizing the encoder network $\phi$ is tricky. Since $\mathbf{z}$ is a discrete latent variable, the gradient from $\log p_\theta(\mathbf{y}, \mathbf{z})$ does not flow through $\mathbf{z}$. Thus, we formulate (2) in a reinforcement learning setting with a one-step Markov Decision Process $(\mathcal{S}, \mathcal{A}, \mathcal{R})$. Under our setting, the state space $\mathcal{S} = \mathcal{D}$; for each state $(\mathbf{x}, \mathbf{y}) \in \mathcal{D}$, the action space $\mathcal{A}_{(\mathbf{x}, \mathbf{y})} = S_{\text{length}(\mathbf{y})}$ with entropy term $\mathcal{H}_{q_\phi}(\cdot|\mathbf{y}, \mathbf{x})$; the reward function $\mathcal{R}((\mathbf{x}, \mathbf{y}), \mathbf{z} \in S_{\text{length}(\mathbf{y})}) = \log p_\theta(\mathbf{y}, \mathbf{z}|\mathbf{x})$. We can then set the optimization objective $L(\phi)$ to be (2). In practice, we find that adding an entropy coefficient $\beta$ and gradually annealing it can speed up the convergence of decoder while still obtaining good autoregressive orders.

To compute $\nabla_\phi L(\phi)$, we derive the policy gradient with baseline formulation (Sutton et al., 2000):

$$\nabla_\phi L(\phi) = \mathbb{E}_{(\mathbf{x}, \mathbf{y}) \sim \mathcal{D}} \left[ \mathbb{E}_{\mathbf{z} \sim q_\phi} \left[ \nabla_\phi \log q_\phi(\mathbf{z}|\mathbf{y}, \mathbf{x})(\log p_\theta(\mathbf{y}, \mathbf{z}|\mathbf{x}) - b(\mathbf{y}, \mathbf{x})) \right] + \beta \nabla_\phi \mathcal{H}_{q_\phi} \right] \quad (3)$$

where $b(\mathbf{y}, \mathbf{x})$ is the baseline function independent of action $\mathbf{z}$. The reason we use a state-dependent baseline $b(\mathbf{y}, \mathbf{x})$ instead of a global baseline $b$ is that the the length of $\mathbf{y}$ can have a wide range, causing significant reward scale difference. In particular, we set $b(\mathbf{y}, \mathbf{x}) = \mathbb{E}_{\mathbf{z} \sim q_\phi} \left[ \log p_\theta(\mathbf{y}, \mathbf{z}_i|\mathbf{x}) \right]$. If we sample $K \geq 2$ latents for each $\mathbf{y}$, then we can use its Monte-Carlo estimate $\frac{1}{K} \sum_{i=1}^{K} \log p_\theta(\mathbf{y}, \mathbf{z}_i|\mathbf{x})$.

Since we use policy gradient to optimize $\phi$, we still need a closed form for the distribution $q_\phi(\mathbf{z}|\mathbf{y}, \mathbf{x})$. Before we proceed, we define $\mathcal{P}_{n \times n}$ as the set of $n \times n$ permutation matrices, where exactly one entry in each row and column is 1 and all other entries are 0; $\mathcal{B}_{n \times n}$ as the set of $n \times n$ doubly stochastic matrices, i.e. non-negative matrices whose sum of entries in each row and in each column equals 1; $\mathbb{R}_{n \times n}^{+}$ as the set of non-negative $n \times n$ matrices. Note that we have the relationship $\mathcal{P}_{n \times n} \subset \mathcal{B}_{n \times n} \subset \mathbb{R}_{n \times n}^{+}$.

To obtain $q_\phi(\mathbf{z}|\mathbf{y}, \mathbf{x})$, we first write $\mathbf{z}$ in two-dimensional form. For each $\mathbf{z} \in S_n$, let $f_n(\mathbf{z}) \in \mathcal{P}_{n \times n}$ be constructed such that $f_n(\mathbf{z})_i = \text{one\_hot}(z_i)$, where $f_n(\mathbf{z})_i$ is the $i$-th row of $f_n(\mathbf{z})$. Thus $f_n$ is a natural bijection from $S_n$ to $\mathcal{P}_{n \times n}$, and we can rewrite $q_\phi$ as a distribution over $\mathcal{P}_{n \times n}$ such that $q_\phi(f_n(\mathbf{z})|\mathbf{y}, \mathbf{x}) = q_\phi(\mathbf{z}|\mathbf{y}, \mathbf{x})$.

Next, we need to model the distribution of $q_\phi(\cdot|\mathbf{y}, \mathbf{x})$. Inspired by (Mena et al., 2018), we model $q_\phi(\cdot|\mathbf{y}, \mathbf{x})$ as a Gumbel-Matching distribution $\mathcal{G.M.}(X)$ over $\mathcal{P}_{n \times n}$, where $X = \phi(\mathbf{y}, \mathbf{x}) \in \mathbb{R}^{n \times n}$ is the output of $\phi$. Then for $P \in \mathcal{P}_{n \times n}$,

$$q_\phi(\mathbf{z}|\mathbf{y}, \mathbf{x}) = q_\phi(f_n^{-1}(P)|\mathbf{y}, \mathbf{x}) = q_\phi(P|\mathbf{y}, \mathbf{x}) \propto \exp \langle X, P \rangle_F \quad (4)$$

where $\langle X, P \rangle_F = \text{trace}(X^T P)$ is the Frobenius inner product of $X$ and $P$. To obtain samples in $\mathcal{P}_{n \times n}$ from the Gumbel-Matching distribution, (Mena et al., 2018) relaxes $\mathcal{P}_{n \times n}$ to $\mathcal{B}_{n \times n}$ by defining the Gumbel-Sinkhorn distribution $\mathcal{G.S.}(X, \tau) : \tau > 0$ over $\mathcal{B}_{n \times n}$, and proves that

$\mathcal{G.S.}(X, \tau)$ converges almost surely to $\mathcal{G.M.}(X)$ as $\tau \to 0^+$. Therefore, to approximately sample from $\mathcal{G.M.}(X)$, we first sample from $\mathcal{G.S.}(X, \tau)$, then apply Hungarian algorithm (Munkres, 1957) to obtain $P \in \mathcal{G.M.}(X)$. Further details are presented in Appendix A.

The Gumbel-Matching distribution allows us to obtain the numerator for the closed form of $q_\phi(\mathbf{z}|\mathbf{y}, \mathbf{x}) = q_\phi(f_n^{-1}(P)|\mathbf{y}, \mathbf{x})$, which equals $\exp \langle X, P \rangle_F$. However, the denominator is intractable to compute and equals $\sum_{P \in \mathcal{P}_{n \times n}} \exp \langle X, P \rangle_F$. Upon further examination, we can express it as $\text{perm}(\exp(X))$, the matrix permanent of $\exp(X)$, and approximate it using $\text{perm}_B(\exp(X))$, its Bethe permanent. We present details about matrix permanent and Bethe permanent along with the proof that the denominator of $q_\phi(\cdot|\mathbf{y}, \mathbf{x})$ equals $\text{perm}(\exp(X))$ in Appendix B.

After we approximate $q_\phi$, we can now optimize $\phi$ using the policy gradient in (3). We present a diagram of our architecture in Figure 1, and a pseudocode of our algorithm in Algorithm 1. Note that even though latent space $S_n$ is very large and contains $n!$ permutations, in practice, if $p_\theta(\mathbf{y}, \mathbf{z}^*|\mathbf{x}) \geq p_\theta(\mathbf{y}, \mathbf{z}|\mathbf{x}) \ \forall \mathbf{z} \in S_n$, then $p_\theta(\mathbf{y}, \mathbf{z}|\mathbf{x})$ tends to increase as the edit distance between $\mathbf{z}$ and $\mathbf{z}^*$ decreases. Therefore, $\phi$ does not need to search over the entire latent to obtain good permutations, making variational inference over $S_n$ feasible.

## 5 Experiments

**Encoder and Decoder Architecture.** We present encoder and decoder architectures for *Variational Order Inference* on conditional sequence generation tasks, which we focus on in this work. Note that Algorithm 1 is also applicable to unconditional sequence generation domains, such as image generation, through different encoder and decoder architectures. We leave this for future work.

For decoder $\theta$, we use the Transformer-InDIGO (Gu et al., 2019a) architecture, which builds on Transformer with relative position representations (Shaw et al., 2018) to allow sequence generation through insertion operations. Note that "encoder" and "decoder" in this section refer to the two networks $\phi$ and $\theta$ in Algorithm 1, respectively, instead of Transformer's encoder and decoder. Also, we obtain orderings through the output of encoder instead of through Searched Adaptive Order (SAO). To our best effort, we were unable to obtain the official implementation of Transformer-InDIGO, so we reimplemented the algorithm based on the paper's descriptions.

As a side note, rather than generating $\mathbf{y}$ autoregressively in $2n$ steps as done in Transformer-InDIGO, it is possible to use a non-autoregressive decoder instead and improve the decoding speed. There have recently been many works on non-autoregressive conditional sequence generation (Gu et al., 2018; 2019b; Ma et al., 2019; Bao et al., 2019). To train a non-autoregressive decoder Transformer, we can incorporate the ordering information generated by our encoder network into the decoder Transformer's encoder latent output. We leave this for future work.

For encoder $\phi$, we adopt the Transformer (Vaswani et al., 2017) architecture. Note that our encoder generates latents based on the entire ground truth target sequence $\mathbf{y}$. Therefore, it does not need to mask out subsequent positions during attention. We also experiment with different position embedding schemes (see Section 7) and find that Transformer-XL's (Dai et al., 2019) relative positional encoding performs the best, so we replace the sinusoid encoding in the original Transformer.

**Tasks.** We evaluate our approach on challenging sequence generation tasks: natural language to code generation (NL2Code) (Ling et al., 2016), image captioning, text summarization, and machine translation. For NL2Code, we use Django (Oda et al., 2015). For image captioning, we use COCO 2017 (Lin et al., 2015). For text summarization, we use English Gigaword (Graff et al., 2003; Rush et al., 2015). For machine translation, we use WMT16 Romanian-English (Ro-En).

**Baselines.** We compare our approach with several pre-defined fixed orders: Left-to-Right (L2R) (Wu et al., 2018), Common-First (Common) (Ford et al., 2018a), Rare-First (Rare) (Ford et al., 2018a), and Random-Ordering (Random). Here, Common-First order is defined as generating words with ordering determined by their relative frequency from high to low; Rare-First order is defined as the reverse of Common-First order; and Random-Ordering is defined as training with a randomly sampled order for each sample at each time step.

**Preprocessing.** For Django, we adopt the same preprocessing steps as described in (Gu et al., 2019a), and we use all unique words as the vocabulary. For MS-COCO, we find that the baseline in Gu et al. (2019a) is much lower than commonly used in the vision and language community. There-

fore, instead of using Resnet-18, we use the pretrained Faster-RCNN checkpoint using a ResNet-50 FPN backbone provided by TorchVision to extract 512-dimensional feature vectors for each object detection. To make our model spatially-aware, we also concatenate the bounding box coordinates for every detection before feeding into our Transformers' encoder. For Gigaword and WMT, we learn 32k byte-pair encoding (BPE, Sennrich et al. (2016)) on tokenized data.

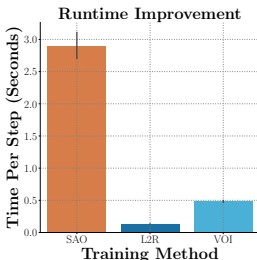
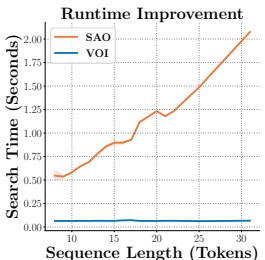
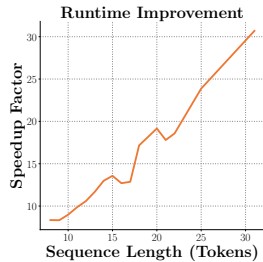

Figure 2: **Runtime performance improvement.** We compare the runtime performance of VOI ($K = 4$) with SAO on a single Tesla P100 GPU, in terms of time per training iteration and ordering search time. VOI outputs latent orderings in a single forward pass, and we observe a significant runtime improvement over SAO that searches orderings sequentially. The speedup factor linearly increases with respect to the sequence length.

| Order | MS-COCO | | | | Django | | Gigaword | | | WMT16 Ro-En | | |
|---|---|---|---|---|---|---|---|---|---|---|---|---|
| | BLEU | Meteor | R-L | CIDEr | BLEU | Accuracy | R-1 | R-2 | R-L | BLEU | Meteor | TER |
| InDIGO - SAO [1] | 29.3 | 24.9 | 54.5 | 92.9 | 42.6 | 32.9 | — | — | — | 32.5 | 53.0 | 49.0 |
| Ours - Random | 28.9 | 24.2 | 55.2 | 92.8 | 21.6 | 26.9 | 30.1 | 11.6 | 27.6 | | | |
| Ours - L2R | 30.5 | 25.3 | 54.5 | 95.6 | 40.5 | 33.7 | 35.6 | 17.2 | 33.2 | 32.7 | 54.4 | 50.2 |
| Ours - Common | 28.0 | 24.8 | 55.5 | 90.3 | 37.1 | 29.8 | 33.9 | 15.0 | 31.1 | 27.4 | 50.1 | 53.9 |
| Ours - Rare | 28.1 | 24.5 | 52.9 | 91.4 | 31.1 | 27.9 | 34.1 | 15.2 | 31.3 | 26.0 | 48.5 | 55.1 |
| Ours - VOI | **31.0** | **25.7** | **56.0** | **100.6** | **44.6** | **34.3** | **36.6** | **17.6** | **34.0** | 32.9 | 54.6 | 49.3 |

Table 1: Results of MS-COCO, Django, Gigaword, and WMT with fixed orders (L2R, Random, Common, Rare) as baseline. Here, R-1, R-2, and R-L indicate ROUGE-1, ROUGE-2, and ROUGE-L, respectively. For TER, lower is better; for all other metrics, higher is better. "—" = not reported.

**Training.** For our decoder, we set $d_{model} = 512$, $d_{hidden} = 2048$, 6 layers for both Transformer's encoder and decoder, and 8 attention heads. This is the same model configuration as Transformer-Base (Vaswani et al., 2017) and as described in Gu et al. (2019a). Our encoder also uses the same configuration. For our model trained with *Variational Order Inference*, we sample $K = 4$ latents for each training sample. An ablation on the choices of $K$ is presented in Section 7. For WMT, many previous works on nonsequential orderings (Stern et al., 2019) and nonautoregressive sequence generation (Gu et al., 2019b) have found sequence-level knowledge distillation (Kim & Rush, 2016) helpful. Therefore, we first train the L2R model on the original WMT corpus, then create a new training corpus using beam search. We find that this improves the BLEU of VOI model by about 2. Even though the training set changed, the orderings learned by VOI are very similar to the ones trained on the original corpus. More detailed training processes are described in Appendix C.

During training, our encoder outputs the latent ordering through one single forward pass, and our decoder can predict all tokens with their positions given by the latent ordering in one single forward pass. If we let $N$ denote the batch size, $l$ denote the length of each target sequence, and $d$ denote the size of hidden vector, then one single forward pass of our model has computation complexity $O(NKdl^2)$, while Transformer-InDIGO trained with SAO has complexity $O(Ndl^3)$. Since $K \ll l$ in general, our algorithm has better theoretical computational complexity during training. During evaluation, we only keep the decoder to iteratively generate the next position and token, which is as efficient as any standard fixed-order autoregressive models.

---

[1] For InDIGO-SAO, we report the results on COCO and Django trained using our own implementation. We did not attempt SAO on Gigaword or WMT due to the large dataset sizes, which can take 100 days to train. For WMT, we report the SAO result as in the original paper, and we follow their evaluation scheme (results are case-sensitive). The BLEU scores are obtained through SacreBLEU. Evaluation scripts are open-sourced.

We also empirically compare VOI's runtime with that of SAO and fixed-order baselines (e.g. L2R). We implement SAO as described in Gu et al. (2019a). We test the runtime on a single GPU in order to accurately measure the number of ops required. For training speed per iteration, we use a batch size of 8. For ordering search time, we use a batch size of 1 to avoid padding tokens in the input for accurate measure. We observe that VOI is significantly faster than SAO, which searches orderings sequentially. In practice, as we distribute VOI across more GPUs, the $K$ factor in the runtime is effectively divided by the number of GPUs used (if we ignore the parallelization overhead), so we can achieve further speedups.

**Results.** We compare VOI against predefined orderings along with Transformer-InDIGO trained with SAO in Table 1. The metrics we used include BLEU-4 (Papineni et al., 2002), Meteor (Denkowski & Lavie, 2014), Rouge (Lin, 2004), CIDEr (Vedantam et al., 2015), and TER (Snover et al., 2006). The "accuracy" reported for Django is defined as the percentage of perfect matches in code generation. Our results illustrate consistently better performance across fixed orderings. Most notably, CIDEr for MS-COCO, BLEU for Django, and Rouge-1 for Gigaword reveal the largest improvements in performance.

## 6   ORDER ANALYSIS

In this section, we analyze the generation orders learned by *Variational Order Inference* on a macro level by comparing the similarity of our learned orders with predefined orders defined in Section 5, and on a micro level, by inspecting when the model generates certain *types* of tokens.

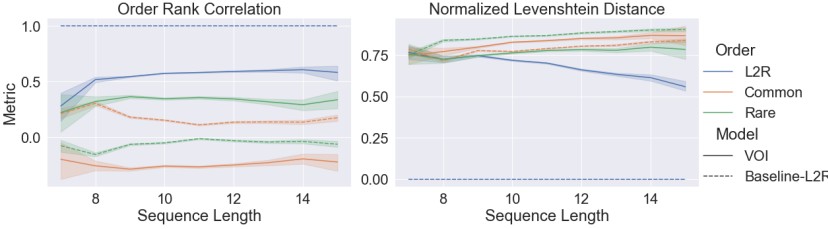

Figure 3: **Global statistics for learned orders.** We compare metrics as a function of the sequence length of generated captions on the COCO 2017 validation set. On the left, we compare orders learned with *Variational Order Inference* to a set of predefined orders (solid lines) using *Order Rank Correlation*. As a reference, we provide the *Order Rank Correlation* between L2R and the same set of predefined orders (dashed lines). In the right plot, with identical setup, we measure *Normalized Levenshtein Distance*. We observe that *Variational Order Inference* favors left-to-right decoding above the other predefined orders—this corresponds to the blue lines. However, with a max *Order Rank Correlation* of 0.6, it appears left-to-right is not a perfect explanation. The comparably high *Order Rank Correlation* of 0.3 with rare-tokens-first order suggests a complex strategy.

### 6.1   UNDERSTANDING THE MODEL GLOBALLY

We find that prior work (Gu et al., 2019a; Welleck et al., 2019a; Gu et al., 2018) tends to study autoregressive orders by evaluating performance on validation sets, and by visualizing the model's generation steps. We provide similar visualizations in Appendix F.3. However, this does not merit a quantitative understanding of the *strategy* that was learned. We address this limitation by introducing methodology to quantitatively study decoding strategies learned by non-monotonic autoregressive models. We introduce *Normalized Levenshtein Distance* and *Order Rank Correlation*, to measure similarity between decoding strategies. Given two generation orders $\mathbf{w}, \mathbf{z} \in S_n$ of the same sequence $\mathbf{y}$, where $n$ is the length of $\mathbf{y}$, we define the *Normalized Levenshtein Distance*.

$$\mathcal{D}_{NLD}(\mathbf{w}, \mathbf{z}) = \operatorname{lev}(\mathbf{w}, \mathbf{z})/n \tag{5}$$

$$\operatorname{lev}(\mathbf{w}, \mathbf{z}) = 1 + \min\{\operatorname{lev}(\mathbf{w}_{1:}, \mathbf{z}), \operatorname{lev}(\mathbf{w}, \mathbf{z}_{1:}), \operatorname{lev}(\mathbf{w}_{1:}, \mathbf{z}_{1:})\} \tag{6}$$

The function $\operatorname{lev}(\mathbf{w}, \mathbf{z})$ is the Levenshtein distance, and $z_{1:}$ removes the first element of $z$. This metric has the property that a distance of $0$ implies that two orders $\mathbf{w}$ and $\mathbf{z}$ are the same, while a distance of $1$ implies that the same tokens appear in distant locations in $\mathbf{w}$ and $\mathbf{z}$. Our second metric *Order Rank Correlation*, is the Spearman's rank correlation coefficient between $\mathbf{w}$ and $\mathbf{z}$.

$$\mathcal{D}_{ORC}(\mathbf{w}, \mathbf{z}) = 1 - 6 \cdot \sum_{i=0}^{n}(\mathbf{w}_i - \mathbf{z}_i)/(n^3 - n) \tag{7}$$

A correlation of 1 implies that **w** and **z** are the same; a correlation of −1 implies that **w** and **z** are reversed; and a correlation of 0 implies that **w** and **z** are not correlated. In Figure 3, we apply these metrics to analyze our models learnt through *Variational Order Inference* .

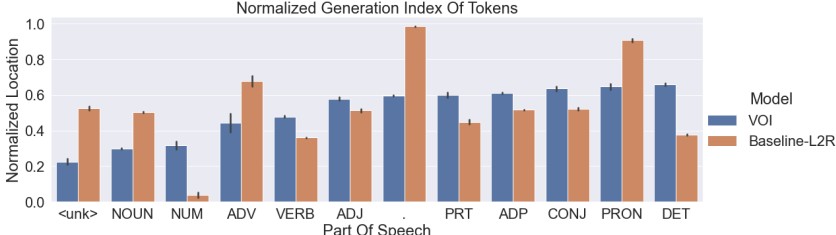

Figure 4: **Local statistics for learned orders.** In this figure, we evaluate the normalized generation indices for different parts of speech in model-predicted captions on the COCO 2017 validation set. The normalized generation index is defined as the absolute generation index of a particular token, divided by the final length of predicted sequence. The parts of speech (details in Appendix E) are sorted in ascending order of their average normalized location. We observe that *modifier* tokens, such as "the", tend to be decoded last, while *descriptive* tokens, such as nouns and verbs, tend to be decoded first.

**Discussion.** The experiment in Figure 3 confirms our model's behavior is not well explained by predefined orders. Interestingly, as the generated sequences increase in length, the *Normalized Levenshtein Distance* decreases, reaching a final value of 0.57, indicating that approximately half of the tokens are already arranged according to a left-to-right generation order. However, the *Order Rank Correlation* barely increases, so we can infer that while individual tokens are close to their left-to-right generation index, their relative ordering is not preserved. Our hypothesis is that certain phrases are generated from left-to-right, but their arrangement follows a *best-first* strategy.

## 6.2 UNDERSTANDING THE MODEL LOCALLY

To complement the study of our model at a global level, we perform a similar study on the micro token level. Our hope is that a per-token metric can help us understand if and when our *Variational Order Inference* is adaptively choosing between left-to-right and rare-first order. We also hope to evaluate our hypothesis that *Variational Order Inference* is following a *best-first* strategy.

**Discussion.** The experiment in Figure 4 demonstrates that *Variational Order Inference* prefers decoding *descriptive* tokens first—such as nouns, numerals, adverbs, verbs, and adjectives. In addition, the unknown part of speech is typically decoded first, and we find this typically corresponds to special tokens such as proper names. Our model appears to capture the *salient* content first, which is illustrated by nouns ranking second in the generation order statistics. For image captioning, nouns typically correspond to focal objects, which suggests our model has an object-detection phase. Evidence of this phase supports our previous hypothesis that a *best-first* strategy is learned.

## 6.3 UNDERSTANDING THE MODEL VIA PERTURBATIONS

In this section, we study the question: to what extent is the generation order learned by *Variational Order Inference* dependent on the content of the conditioning variable **x**? This question is important because simply knowing that our model has learned a *best-first* does not illuminate whether that strategy depends only on the target tokens **y** being generated, or if it also depends on the content of **x**. An adaptive generation order should depend on both.

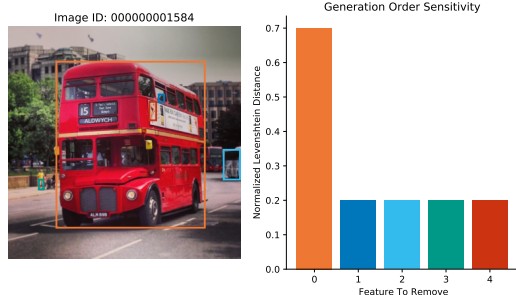

**Discussion.** In this experiment, we first obtain a sequence **y** generated by our VOI given the source image **x**. We then freeze **y**, which allows the model to infer a new generation order for **y** when

different features of **x** are removed. Figure 6.3 shows that for a particular case, removing a single region-feature (feature number 0, which corresponds to the bus) from **x** changes the model-predicted generation order by as much as 0.7 *Normalized Levenshtein Distance*. These results confirm that our model appears to learn an *adaptive* strategy, which depends on both the tokens **y** being generated and the content of the conditioning variable **x**, which is an image in this experiment.

## 7 ABLATION STUDIES

In Section 5, we introduced the specific encoder and decoder architectures we use for conditional sequence generation tasks. In this section, we present ablation studies to support the architecture design of our encoder and modeling $q_\phi$ with Gumbel-Matching distribution.

We consider 4 different positional encoding schemes for the encoder Transformer $\phi$: the sinusoid encoding in the original Trans-

Table 2: Normalized Levenshtein Distance between the ordering learnt by the encoder and the ground truth ordering, under different positional encodings (enc) and modeling distributions of $q_\phi$ (distrib).

| Enc \ Distrib | Gumbel-Matching | Plackett-Luce |
|---|---|---|
| Sinusoid | 0.40 | 0.62 |
| Sinusoid + Pos Attn | 0.42 | 0.58 |
| Relative | 0.38 | 0.53 |
| XL-Relative | **0.25** | 0.57 |

former (Vaswani et al., 2017), the sinusoid encoding with positional attention module (Gu et al., 2018), the relative positional encoding in Shaw et al. (2018), and the relative positional encoding proposed in Transformer-XL (Dai et al., 2019). Besides modeling $q_\phi(\cdot|\mathbf{x}, \mathbf{y})$ as Gumbel-Matching distribution and using Bethe permanent to approximate its denominator, we also consider modeling using Plackett-Luce distribution (Plackett, 1975; Luce, 1959) and sample using techniques recently proposed in Grover et al. (2019). Plackett-Luce distribution has tractable density, so we can compute the exact $q_\phi$ efficiently without using approximation techniques.

To analyze the encoder's ability to learn autoregressive orderings, we first train a decoder with Common-First order on one batch of MS-COCO until it perfectly generates each sentence. We then fix the decoder and initialize an encoder. We train the encoder for 15k gradient steps using the procedure in Algorithm 1 to recover the ground truth Common-First order, and we report the final Normalized Levenshtein Distance against the ground truth in Table 2. We observe that modeling $q_\phi$ with Gumbel-Matching distribution significantly outperforms modeling with Plackett-Luce, despite the former requiring denominator approximation. We also observe that under Gumbel-Matching modeling distribution, the relative position encoding in Transformer-XL significantly outperforms other encoding schemes. Thus we combine these two techniques in our architecture design.

In addition, we analyze how the choice of $K$, the number of latents per training sample, affects model performance. We use the same setting as above and apply Transformer-XL relative position encoding, and we report the results in Table 3. We observe that the encoder more accurately fits to the ground

Table 3: Normalized Levenshtein Distance between the encoder ordering and the ground truth with respect to the choice of $K$.

| $K$ | 2 | 3 | 4 | 10 | 20 |
|---|---|---|---|---|---|
| $\mathcal{D}_{NLD}$ | 0.31 | 0.28 | 0.25 | 0.21 | 0.21 |

truth order as $K$ increases, until a value of around 10. Since a very large $K$ can slow the model down while only bringing marginal improvement, a choice of $K$ from 4 to 10 is sufficient.

## 8 CONCLUSION

We propose, to our best knowledge, the first unsupervised learner that learns high-quality autoregressive orders through fully-parallelizable end-to-end training without domain-specific tuning. We propose a procedure named *Variational Order Inference* that uses the Variational Lower Bound with the space of autoregressive orderings as latent. Building on techniques in combinatorical optimization, we develop a practical policy gradient algorithm to optimize the encoder of the variational objective, and we propose an encoder architecture that conditions on training examples to output autoregressive orders. Empirical results demonstrate that our model is capable of discovering autoregressive orders that are competitive with or even better than fixed and predefined orders. In addition, the global and local analysis of the orderings learned through *Variational Order Inference* suggest that they resemble a type of *best-first* generation order, characterized by prioritizing the generation of *descriptive* tokens and deprioritizing the generation of *modifier* tokens.

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

# APPENDIX

## A   GUMBEL-MATCHING DISTRIBUTION AND ITS SAMPLING

In Section 4, We model the distribution of $q_\phi(\cdot|\mathbf{y}, \mathbf{x})$ as a Gumbel-Matching distribution $\mathcal{G}.\mathcal{M}.(X)$ over $\mathcal{P}_{n \times n}$, where $X = \phi(\mathbf{y}, \mathbf{x}) \in \mathbb{R}^{n \times n}$ is the latent output.

To obtain samples in $\mathcal{P}_{n \times n}$ from the Gumbel-Matching distribution, Mena et al. (2018) relaxes $\mathcal{P}_{n \times n}$ to $\mathcal{B}_{n \times n}$ by defining the Gumbel-Sinkhorn distribution $\mathcal{G}.\mathcal{S}.(X, \tau) : \tau > 0$ over $\mathcal{B}_{n \times n}$. Here we reproduce the following definitions and theorems with similar notations from Sinkhorn (1964) and Mena et al. (2018):

**Definition A.1.** *Let $X \in \mathbb{R}_{n \times n}$ and $A \in \mathbb{R}_{n \times n}^+$. The **Sinkhorn Operator** $S$ is defined as*

$$\mathcal{T}_r(A) = A \oslash (A\mathbf{1}_n\mathbf{1}_n^T) \tag{8}$$

$$\mathcal{T}_c(A) = A \oslash (\mathbf{1}_n\mathbf{1}_n^T A) \tag{9}$$

$$\mathcal{T}(A) = \mathcal{T}_r(\mathcal{T}_c(A)) \tag{10}$$

$$S(X) = \lim_{n \to \infty} \mathcal{T}^n(\exp(X)) \tag{11}$$

Here, $\oslash$ is the element-wise division between two matrices, and $\mathcal{T}_r$ and $\mathcal{T}_c$ are row and column normalizations of a non-negative matrix, respectively. Therefore, iteratively applying $\mathcal{T}$ is equivalent to iteratively normalizing a non-negative matrix by column and row.

**Theorem A.2.** (Sinkhorn, 1964) *The range of $S$ is $\mathcal{B}_{n \times n}$.*

**Theorem A.3.** (Mena et al., 2018) *Let $X \in \mathbb{R}_{n \times n}, \tau > 0$. The **Gumbel-Sinkhorn distribution** $\mathcal{G}.\mathcal{S}.(X, \tau)$ is defined as follows:*

$$\mathcal{G}.\mathcal{S}.(X, \tau) = S(\frac{X + \epsilon}{\tau}) \tag{12}$$

*where $\epsilon$ is a matrix of i.i.d. standard Gumbel noise. Moreover, $\mathcal{G}.\mathcal{S}.(X, \tau)$ converges almost surely to $\mathcal{G}.\mathcal{M}.(X)$ as $\tau \to 0^+$.*

To approximately sample from $\mathcal{G}.\mathcal{M}.(X)$, we first sample from $\mathcal{G}.\mathcal{S}.(X, \tau)$. Even though theoretically, $\mathcal{T}$ needs to be applied infinite number of times to obtain a matrix in $\mathcal{B}_{n \times n}$, Mena et al. (2018) reports that 20 iterations of $\mathcal{T}$ are enough in practice. We find that in our experiments, 20 iterations are not enough to obtain a matrix in $\mathcal{B}_{n \times n}$, but $100 - 200$ iterations are enough. After we obtain the matrix in $\mathcal{B}_{n \times n}$, we apply Hungarian algorithm (Munkres, 1957) to obtain $P \in \mathcal{G}.\mathcal{M}.(X)$.

Finally, we need to calculate the entropy term $\mathcal{H}_{q_\phi}$ in $L_\phi$ in Equation 3. This can be approximated using the technique in Appendix B.3 of Mena et al. (2018).

## B   MATRIX PERMANENT AND ITS APPROXIMATION WITH BETHE PERMANENT

In this section, we present details about matrix permanent and bethe permanent, which we use as an approximation to the denominator of $q_\phi(\cdot|\mathbf{y}, \mathbf{x})$.

**Definition B.1.** *Let $A \in \mathbb{R}_{n \times n}$. The **permanent** of $A$ is defined as follows:*

$$perm(A) = \sum_{\sigma \in S_n} \prod_{i=1}^{n} A_{i, \sigma_i} \tag{13}$$

**Theorem B.2.** *The denominator of $q_\phi(\cdot|\mathbf{y}, \mathbf{x})$ equals $perm(\exp(X))$.*

*Proof.*

$$\sum_{P \in \mathcal{P}_{n \times n}} \exp \langle X, P \rangle_F = \sum_{\sigma \in S_n} \exp(\sum_{i=1}^{n} X_{i,\sigma(i)})$$

$$= \sum_{\sigma \in S_n} \prod_{i=1}^{n} (\exp(X))_{i,\sigma(i)}$$

$$= \mathrm{perm}(\exp(X))$$

$\square$

**Definition B.3.** (Vontobel, 2010; Anari & Rezaei, 2019) *Let $A \in \mathbb{R}_{n \times n}^{+}$. The **bethe permanent** of $A$ is defined as follows:*

$$perm_B(A) = \exp\big(\max_{\gamma \in \mathcal{B}_{n \times n}} \sum_{i,j} (\gamma_{i,j} \log A_{i,j} - \gamma_{i,j} \log \gamma_{i,j} + (1 - \gamma_{i,j}) \log (1 - \gamma_{i,j}))\big) \quad (14)$$

**Theorem B.4.** (Anari & Rezaei, 2019) *Let $A \in \mathbb{R}_{n \times n}^{+}$. Then, $\sqrt{2}^{-n} perm(A) \leq perm_B(A) \leq perm(A)$.*

The $\gamma$ in Definition B.3 can be calculated using the message passing algorithm in Lemma 29 of Vontobel (2010). An efficient implementation has recently been introduced in Appendix C of Mena et al. (2020). Therefore, we can use $perm_B(\exp(X))$ to approximate the denominator of $q_\phi(\cdot|\mathbf{y}, \mathbf{x})$, and we can then use policy gradient to compute $\nabla_\phi L(\phi)$ in Equation (3).

## C  DETAILED TRAINING PROCESS AND HYPERPARAMETER SETTINGS

For all experiments, we apply dropout = 0.1 (Srivastava et al., 2014) and label smoothing = 0.1. We apply Adam Optimizer (Kingma & Ba, 2015) with $\beta_1 = 0.99, \beta_2 = 0.999$ for MS-COCO, and $\beta_1 = 0.99, \beta_2 = 0.98$ for all other tasks. For baseline experiments, we use a batch size of 64 for Django and MS-COCO, and 128 for Gigaword and WMT. We decrease the learning rate linearly from 1e-4 to zero. We train the baseline until the performance plateaus.

For our VOI model, we train on Django for a total of 350 epochs (120k gradient steps), MS-COCO for 20 epochs (350k gradient steps), Gigaword for 16 epochs (1M gradient steps), and WMT16 Ro-En for 120 epochs (1.3M gradient steps). We use a batch size of 36 for MS-COCO and Django, 50 for Gigaword, and 54 for WMT. We sample $K = 4$ latents per training sample for the first three datasets, and $K = 3$ for WMT. Due to constraints in computational resource, we were unable to scale WMT to larger batch size and larger $K$. We also did not experiment with larger batch size for COCO, Django, and Gigaword. We leave the investigations of larger batch sizes and larger $K$ for future work.

We set the initial decoder learning rate to be 5e-5 and the encoder learning rate to be 5e-6. We train the VOI encoder and decoder with shared embedding for about the first 15-20% of steps (i.e. 4 epochs for COCO, 50 epochs for Django, 3 epochs for Gigaword, and 20 epochs for WMT). We then separate the embeddings for the rest of the training steps. When the embedding is shared, we set the entropy coefficient $\beta = 0.3$ for all tasks.

After we separate the embeddings, for MS-COCO, we anneal $\beta$ with a log-linear schedule from 0.3 to 0.03. We decrease the learning rate to (3e-5, 3e-6) for the decoder and the encoder respectively after epoch 13, when the encoder starts to sample very similar permutations for a single training data. We observe that training either VOI or the fixed ordering models for too long leads to overfitting. Finetuning VOI with the encoder fixed does not help and causes the performance to slightly drop.

For Django, we set the learning rates to be (3e-5, 3e-6). We log-anneal $\beta$ to 0.03 for the first 90% of steps and then anneal $\beta$ to 0.003 for the rest of the steps. We find that the latter allows the encoder to commit to a single ordering on sequences of longer length and slightly improves performance. We finally fix the encoder and finetune the decoder for 50 epochs with a larger batch size of 64 and learning rate linearly annealing to zero. This finetuning step improves the BLEU score by about 0.6.

For Gigaword and WMT, we add a cosine alignment loss between the decoder and the encoder's embedding matrices to the loss of the encoder. We set the cosine alignment loss coefficient to be

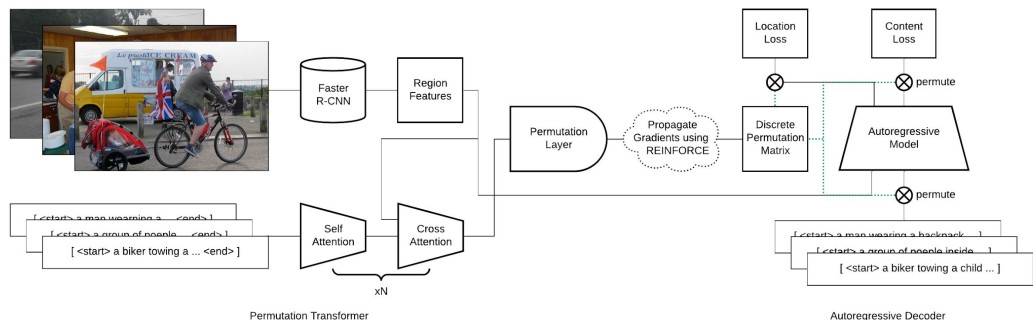

Figure 5: This figure demonstrates our algorithm for an image captioning task. The model on the left is the Permutation Transformer, which maps training examples to permutation matrices. The model on the right is an autoregressive model that learns to predict tokens and positions.

100.0 for Gigaword and 10.0 for WMT. Intuitively, since the Gigaword and WMT vocabularies are much larger than those of MS-COCO and Django, and they contain many rare words, this loss allows the encoder to leverage the semantic information recently-learnt from the decoder to better discover autoregressive orderings.

For Gigaword, we anneal $\beta$ log-linearly from 0.3 to 0.03 in 8 epochs (500k gradient steps). We then fix the encoder and fine-tune the decoder with a batch size of 128 for 5 epochs with learning rate linearly decreasing from 7e-5 to 0. We observe that, compared to COCO and Django, this finetuning step significantly improves VOI's performance and raises the ROUGE score by around 1.5 to 2.0.

For WMT, we anneal $\beta$ log-linearly from 0.3 to 7e-4 in 80 epochs (900k gradient steps). We decrease the learning rates from (5e-5, 5e-6) to (3e-5, 3e-6) at epoch 40 when the encoder starts sampling very similar permutations. We then fix the encoder and finetune the decoder with a batch size of 128 for 20 epochs with learning rate linearly decreasing from 3e-5 to 0. We observe that this finetuning step also significantly benefits VOI's performance and improves the BLEU score by around 1.5 points.

Due to resource constraints, we did not tune our hyperparameters and training schedules very carefully, and we leave the discovery of better training schemes for future work.

## D   EXAMPLE ARCHITECTURE FOR CONDITIONAL SEQUENCE GENERATION

In Section 5, we introduced the specific encoder and decoder architectures used for the conditional sequence generation tasks in our paper. To further illustrate the architecture of *Variational Order Inference* , we present a diagram of the architecture instantiated for COCO 2017.

## E   PARTS OF SPEECH MAPPINGS

The parts of speech used in our Order Analysis section correspond to the NLTK Universal Tagset. In the below table, we provide mappings for the tag identifiers used in our main paper. More information about the specific NLTK tags can be found at the following url: http://www.nltk.org/book/ch05.html.

## F   VISUALIZATIONS OF SEQUENCE GENERATION

### F.1   COCO

We visualize the generation order inferred by *Variational Order Inference* for COCO. Sequences are generated using beam search over both tokens and their insertion positions, using a beam size of 3. Bounding boxes that correspond to region-features calculated using bottom-up attention are superimposed on the image, with an opacity value proportional to the magnitude of their softmax attention value in the final cross-attention layer in the language model.

| Tag | Meaning | English Examples |
|------|-------------------|------------------------------------|
| ADJ | adjective | new, good, high, special, big, local |
| ADP | adposition | on, of, at, with, by, into, under |
| ADV | adverb | really, already, still, early, now |
| CONJ | conjunction | and, or, but, if, while, although |
| DET | determiner, article | the, a, some, most, every, no, which |
| NOUN | noun | year, home, costs, time, Africa |
| NUM | numeral | twenty-four, fourth, 1991, 14:24 |
| PRT | particle | at, on, out, over per, that, up, with |
| PRON | pronoun | he, their, her, its, my, I, us |
| VERB | verb | is, say, told, given, playing, would |
| . | punctuation marks | . , ; ! |
| X | other | ersatz, esprit, dunno, gr8, univeristy |

Table 4: NLTK Universal Tagset.

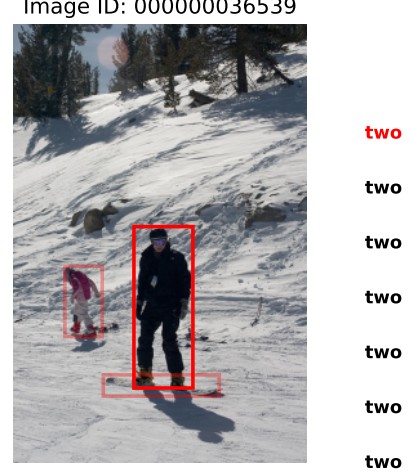

Figure 6: Generation order inferred by **Variational Order Inference**. Without supervision over its generation order, nor a domain-specific initialization, nor a prior to aid learning, the model learns an adaptive strategy that prioritizes object names—in this case, *people* and *snow*.

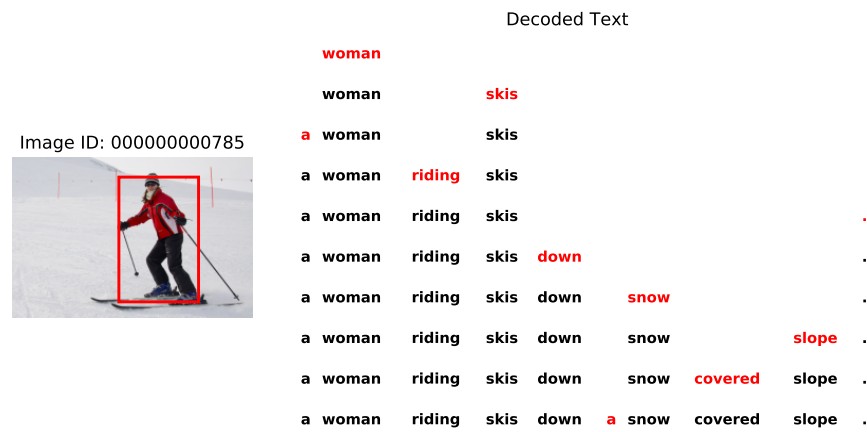

Figure 7: Generation order inferred by **Ours-VOI** for an image from the COCO 2017 validation set with the image identifier **000000000785**.

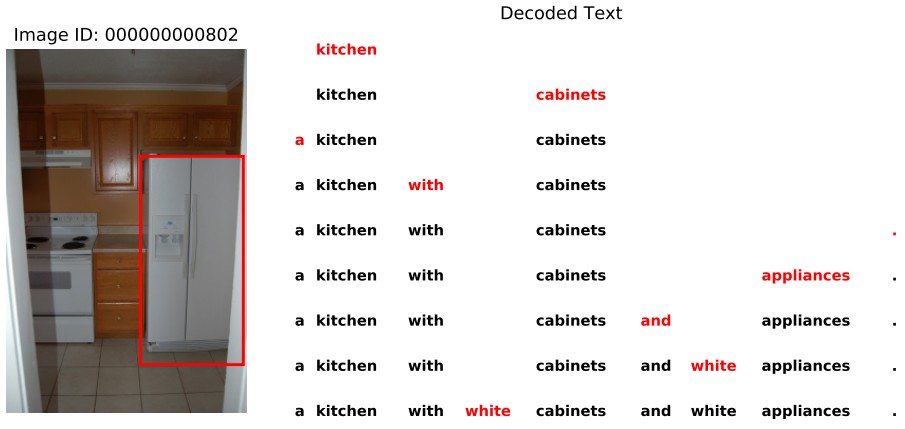

Figure 8: Generation order inferred by **Ours-VOI** for an image from the COCO 2017 validation set with the image identifier **000000000802**.

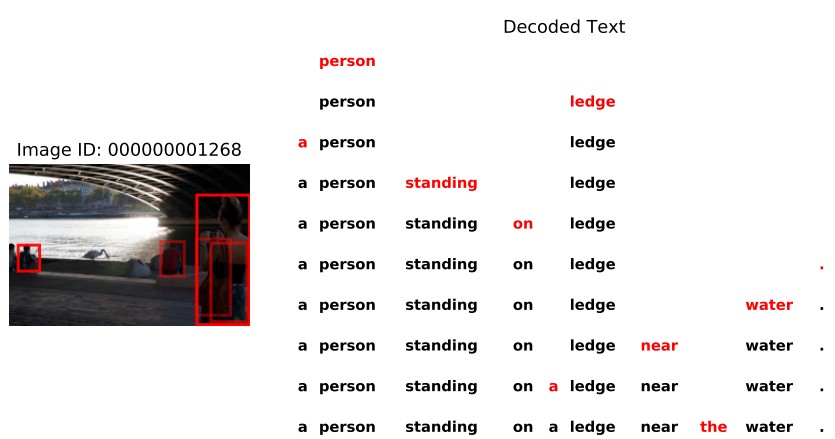

Figure 9: Generation order inferred by **Ours-VOI** for an image from the COCO 2017 validation set with the image identifier **000000001268**.

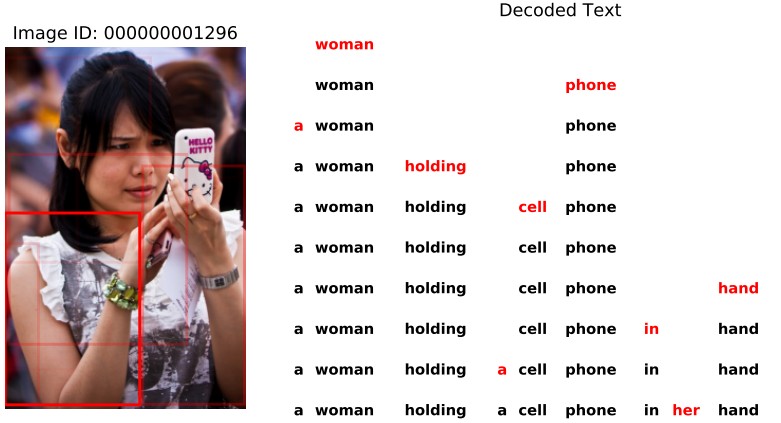

Figure 10: Generation order inferred by **Ours-VOI** for an image from the COCO 2017 validation set with the image identifier **000000001296**.

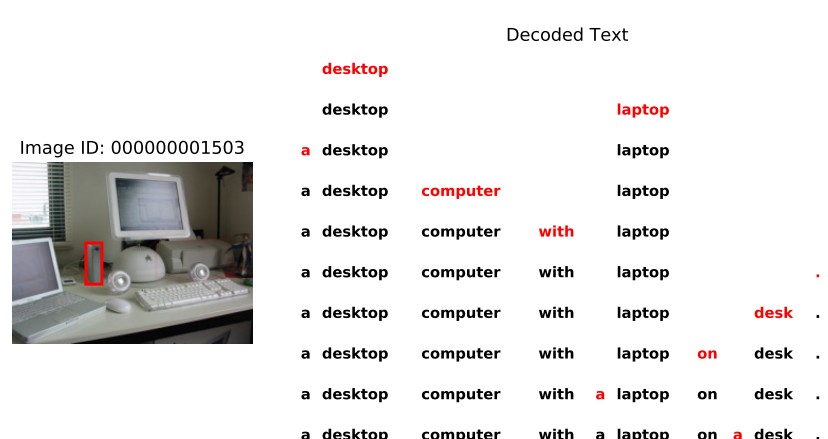

Figure 11: Generation order inferred by **Ours-VOI** for an image from the COCO 2017 validation set with the image identifier **000000001503**.

Decoded Text

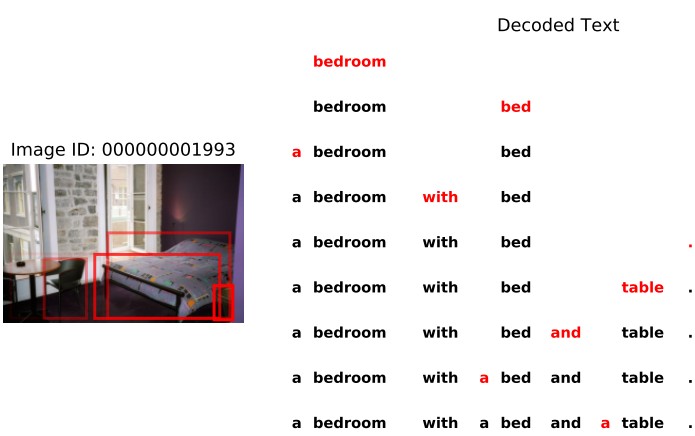

Figure 12: Generation order inferred by **Ours-VOI** for an image from the COCO 2017 validation set with the image identifier **000000001993**.

Decoded Text

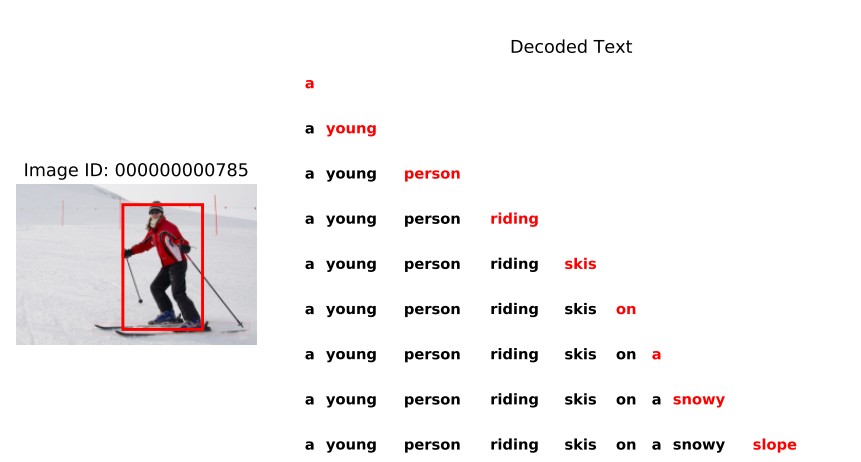

Figure 13: Generation order inferred by **Ours-L2R** for an image from the COCO 2017 validation set with the image identifier **000000000785**.

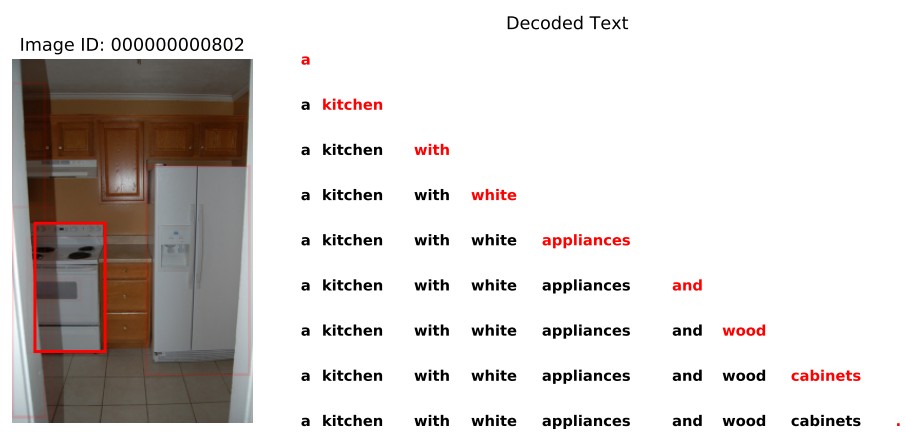

Figure 14: Generation order inferred by **Ours-L2R** for an image from the COCO 2017 validation set with the image identifier **000000000802**.

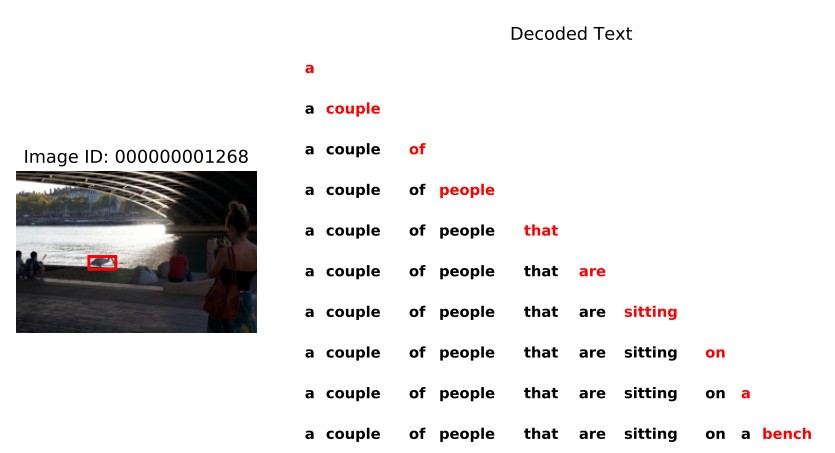

Figure 15: Generation order inferred by **Ours-L2R** for an image from the COCO 2017 validation set with the image identifier **000000001268**.

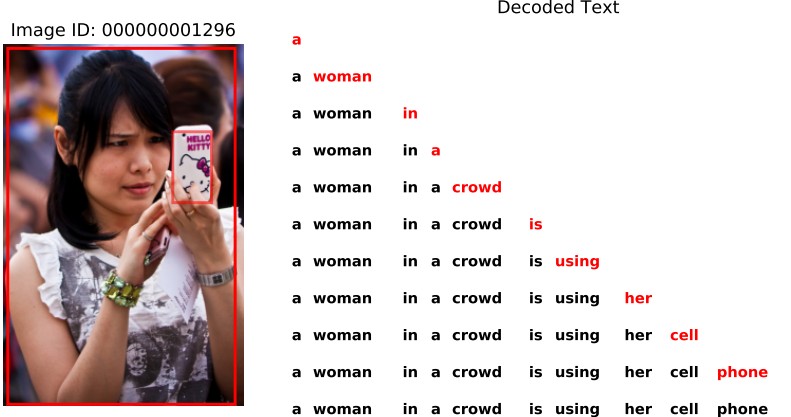

Figure 16: Generation order inferred by **Ours-L2R** for an image from the COCO 2017 validation set with the image identifier **000000001296**.

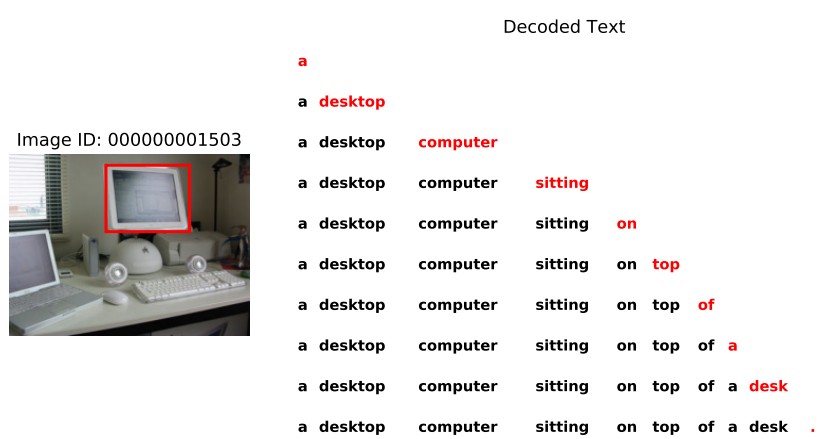

Figure 17: Generation order inferred by **Ours-L2R** for an image from the COCO 2017 validation set with the image identifier **000000001503**.

Decoded Text

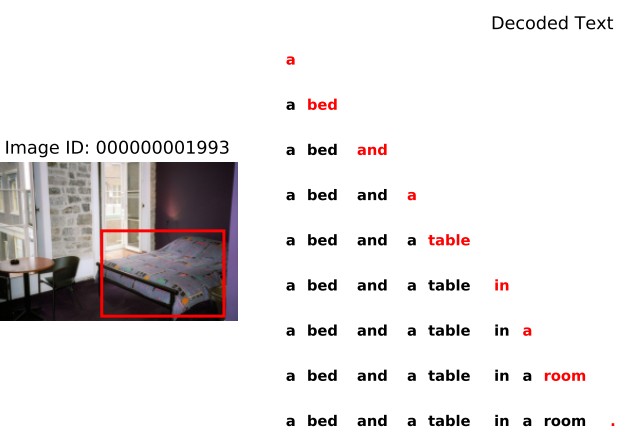

Image ID: 000000001993

a

a **bed**

a bed **and**

a bed and **a**

a bed and a **table**

a bed and a table **in**

a bed and a table in **a**

a bed and a table in a **room**

a bed and a table in a room **.**

Figure 18: Generation order inferred by **Ours-L2R** for an image from the COCO 2017 validation set with the image identifier **000000001993**.

Decoded Text

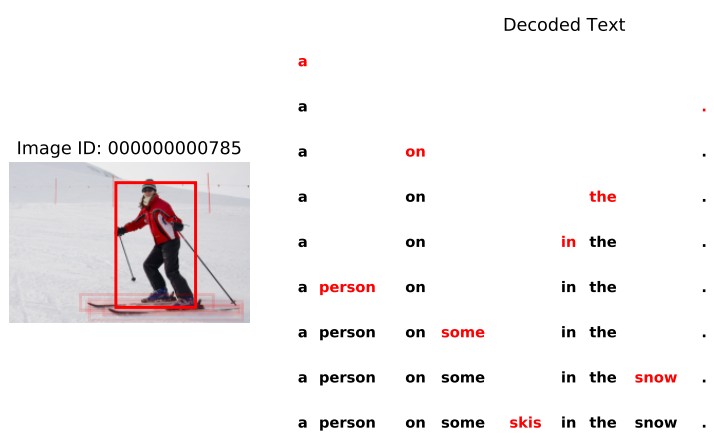

Image ID: 000000000785

a

a **.**

a **on** .

a on **the** .

a on **in** the .

a **person** on in the .

a person on **some** in the .

a person on some in the **snow** .

a person on some **skis** in the snow .

Figure 19: Generation order inferred by **Ours-Common** for an image from the COCO 2017 validation set with the image identifier **000000000785**.

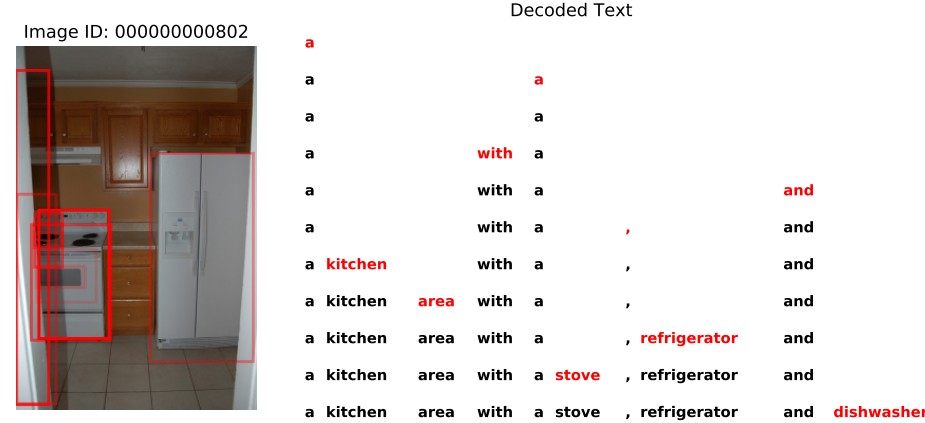

Figure 20: Generation order inferred by **Ours-Common** for an image from the COCO 2017 validation set with the image identifier **000000000802**.

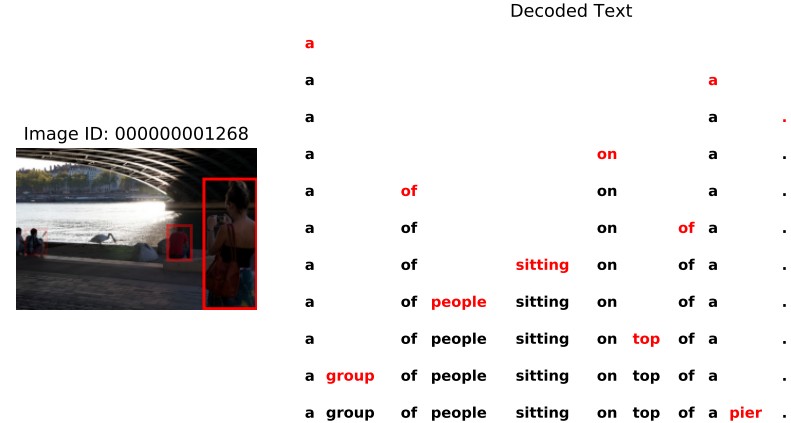

Figure 21: Generation order inferred by **Ours-Common** for an image from the COCO 2017 validation set with the image identifier **000000001268**.

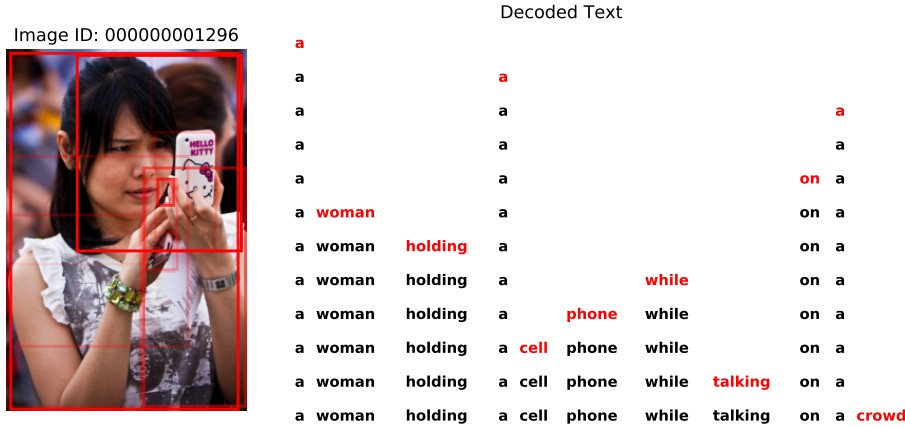

Figure 22: Generation order inferred by **Ours-Common** for an image from the COCO 2017 validation set with the image identifier **000000001296**.

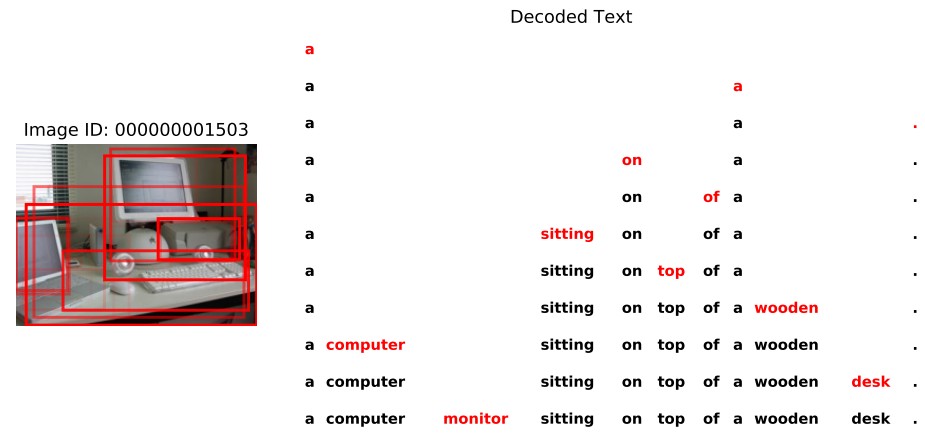

Figure 23: Generation order inferred by **Ours-Common** for an image from the COCO 2017 validation set with the image identifier **000000001503**.

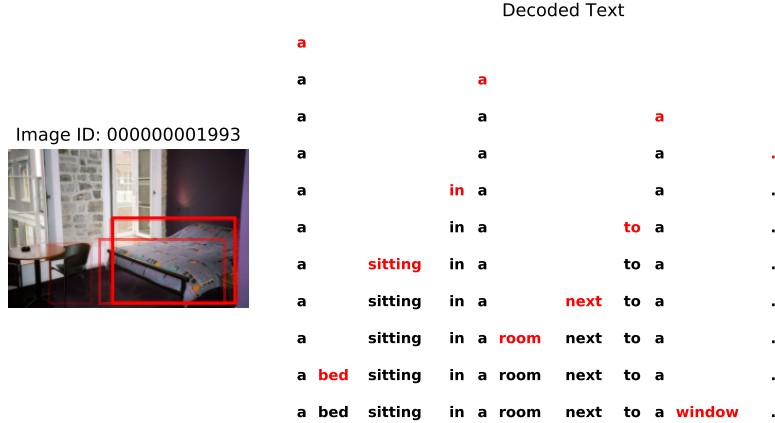

Decoded Text

Image ID: 000000001993

```
a
a                       a
a                       a                           a
a                       a                           a            .
a               in      a                           a            .
a               in      a                   to      a            .
a       sitting in      a                   to      a            .
a       sitting in      a           next    to      a            .
a       sitting in      a   room    next    to      a            .
a bed   sitting in      a   room    next    to      a            .
a bed   sitting in      a   room    next    to      a   window   .
```

Figure 24: Generation order inferred by **Ours-Common** for an image from the COCO 2017 validation set with the image identifier **000000001993**.

Decoded Text

Image ID: 000000000785

```
                                                            slope
                        skis                                slope
                        skis                    covered     slope
                        skis            snow    covered     slope
                riding  skis            snow    covered     slope
                riding  skis    down    snow    covered     slope
        man     riding  skis    down    snow    covered     slope
        man     riding  skis    down    snow    covered     slope   .
a       man     riding  skis    down    snow    covered     slope   .
a       man     riding  skis    down  a snow    covered     slope   .
```

Figure 25: Generation order inferred by **Ours-Rare** for an image from the COCO 2017 validation set with the image identifier **000000000785**.

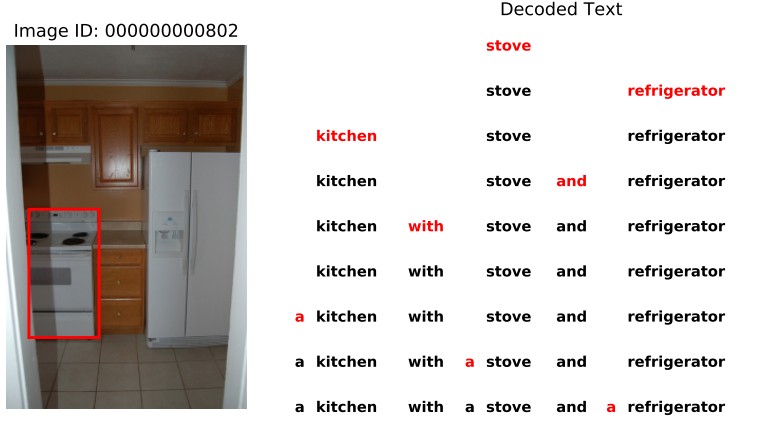

Figure 26: Generation order inferred by **Ours-Rare** for an image from the COCO 2017 validation set with the image identifier **000000000802**.

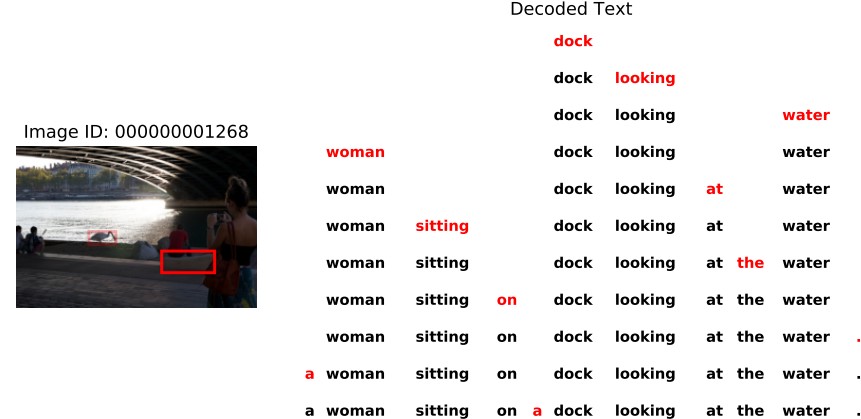

Figure 27: Generation order inferred by **Ours-Rare** for an image from the COCO 2017 validation set with the image identifier **000000001268**.

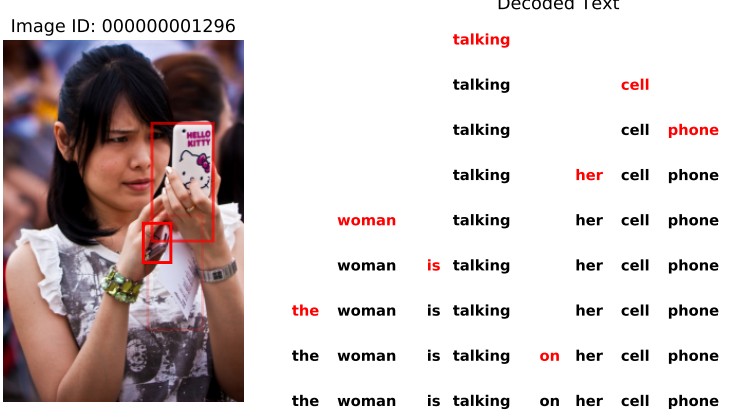

Figure 28: Generation order inferred by **Ours-Rare** for an image from the COCO 2017 validation set with the image identifier **000000001296**.

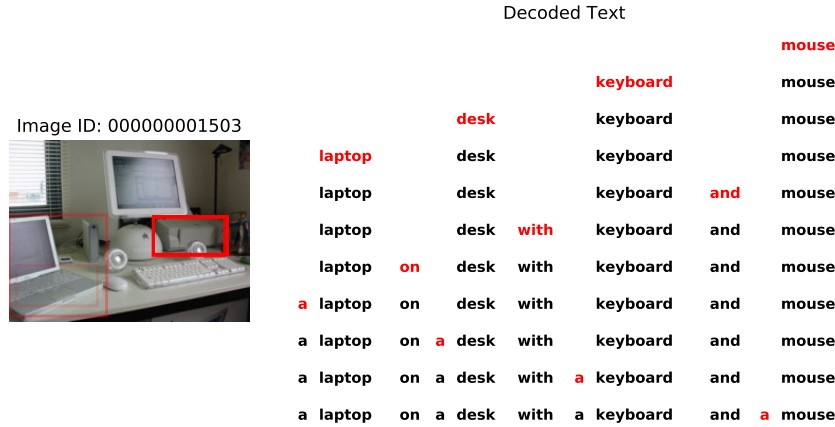

Figure 29: Generation order inferred by **Ours-Rare** for an image from the COCO 2017 validation set with the image identifier **000000001503**.

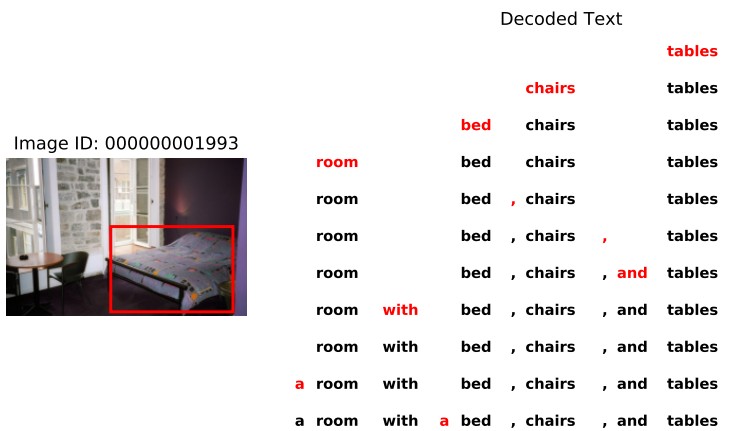

Figure 30: Generation order inferred by **Ours-Rare** for an image from the COCO 2017 validation set with the image identifier **000000001993**.

## F.2 DJANGO

We visualize the latent generation order inferred by Variational Order Inference for Django. Sequences are generated using a beam search over both the tokens and their insertion positions, using a beam size of 3. Text on which the model is conditioned is provided on the left for each example.

Figure 31: Generation order inferred by **Ours-VOI** for a pseudocode sample from the Django natural language to code test set with the sample id **154**.

## F.3 GIGAWORD

We visualize the latent generation order inferred by Variational Order Inference for Gigaword. Sequences are generated using a beam search over both the tokens and their insertion positions, using a beam size of 3. Text on which the model is conditioned is provided on the left for each example.

Conditioned Text                    Decoded Text

|  |  |  |  |  |  |  |  |  |  |  |  |
|---|---|---|---|---|---|---|---|---|---|---|---|
|  |  |  | **i** |  |  |  |  |  |  |  |  |
|  |  |  | i |  | **enumerate** |  |  |  |  |  |  |
|  |  |  | i | **,** | enumerate |  |  |  |  |  |  |
|  |  |  | i | , **arg** | enumerate |  |  |  |  |  |  |
|  |  |  | i | , arg | **in** enumerate |  |  |  |  |  |  |
| for every i and arg in enumerated iterable args , |  |  | i | , arg | in enumerate |  |  | **:pass** |  |  |  |
|  |  |  | i | , arg | in enumerate | **(** |  | :pass |  |  |  |
|  |  |  | i | , arg | in enumerate | ( | **args** | :pass |  |  |  |
|  |  |  | i | , arg | in enumerate | ( | args | **)** :pass |  |  |  |
|  |  | **for** | i | , arg | in enumerate | ( | args | ) :pass |  |  |  |

Figure 32: Generation order inferred by **Ours-VOI** for a pseudocode sample from the Django natural language to code test set with the sample id **431**.

Conditioned Text                    Decoded Text

|  |  |  |  |  |  |  |  |
|---|---|---|---|---|---|---|---|
| | **raise** | | | | | | |
| | raise | **AttributeError** | | | | | |
| | raise | AttributeError | **(** | | | | |
| | raise | AttributeError | ( **'_STR:0_'** | | | | |
| raise an AttributeError with an argument string _STR:0_ , formated with self.name [ self . name ] . | raise | AttributeError | ( '_STR:0_' | **%** | | | |
| | raise | AttributeError | ( '_STR:0_' | % **self** | | | |
| | raise | AttributeError | ( '_STR:0_' | % self | **.** | | |
| | raise | AttributeError | ( '_STR:0_' | % self | . **name** | | |
| | raise | AttributeError | ( '_STR:0_' | % self | . name | **)** | |

Figure 33: Generation order inferred by **Ours-L2R** for a pseudocode sample from the Django natural language to code test set with the sample id **154**.

Conditioned Text                                    Decoded Text

**for**

for  **i**

for  i  **,**

for  i  ,  **arg**

for  i  ,  arg  **in**

**for every i and arg in enumerated
iterable args ,**

for  i  ,  arg  in  **enumerate**

for  i  ,  arg  in  enumerate  **(**

for  i  ,  arg  in  enumerate  (  **args**

for  i  ,  arg  in  enumerate  (  args  **)**

for  i  ,  arg  in  enumerate  (  args  )  **:pass**

Figure 34: Generation order inferred by **Ours-L2R** for a pseudocode sample from the Django natural language to code test set with the sample id **431**.

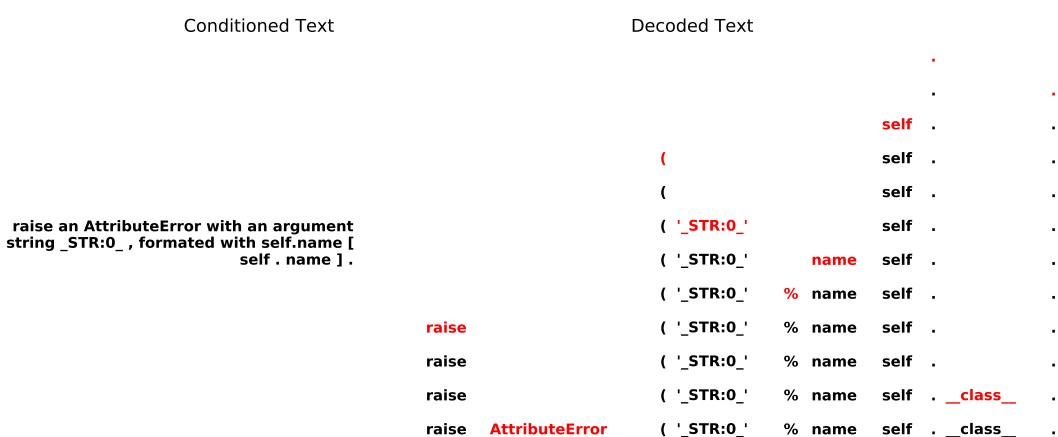

Figure 35: Generation order inferred by **Ours-Common** for a pseudocode sample from the Django natural language to code test set with the sample id **154**.

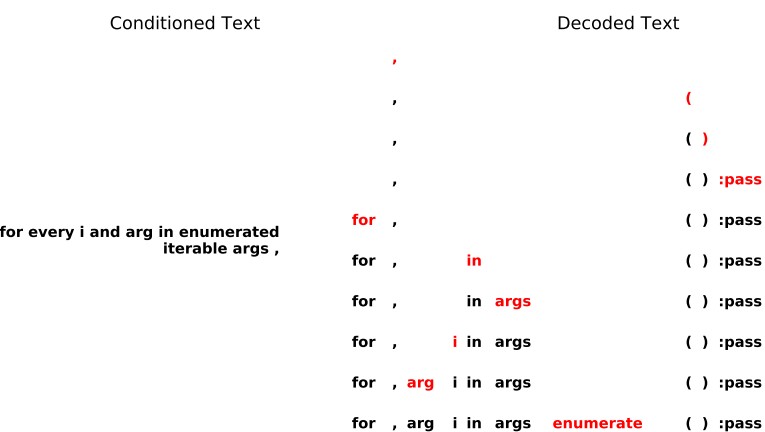

Figure 36: Generation order inferred by **Ours-Common** for a pseudocode sample from the Django natural language to code test set with the sample id **431**.

|  | Conditioned Text |  |  |  |  | Decoded Text |  |  |  |  |  |
|---|---|---|---|---|---|---|---|---|---|---|---|
|  |  | **,** |  |  |  |  |  |  |  |  |  |
|  |  | **,** |  |  |  | **(** |  |  |  |  |  |
|  |  | **,** |  |  |  | **( )** |  |  |  |  |  |
|  |  | **,** |  |  |  | **( )** | **:pass** |  |  |  |  |
| **for every i and arg in enumerated** | **for** | **,** |  |  |  | **( )** | **:pass** |  |  |  |  |
| **iterable args ,** | **for** | **,** |  | **in** |  | **( )** | **:pass** |  |  |  |  |
|  | **for** | **,** |  | **in** | **args** | **( )** | **:pass** |  |  |  |  |
|  | **for** | **,** | **i** | **in** | **args** | **( )** | **:pass** |  |  |  |  |
|  | **for** | **,** | **arg** | **i** | **in** | **args** | **( )** | **:pass** |  |  |  |
|  | **for** | **,** | **arg** | **i** | **in** | **args** | **enumerate** | **( )** | **:pass** |  |  |

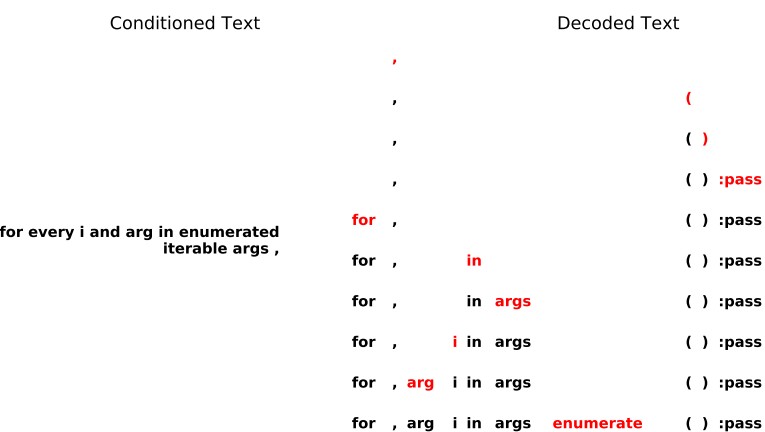

Figure 37: Generation order inferred by **Ours-Rare** for a pseudocode sample from the Django natural language to code test set with the sample id **154**.

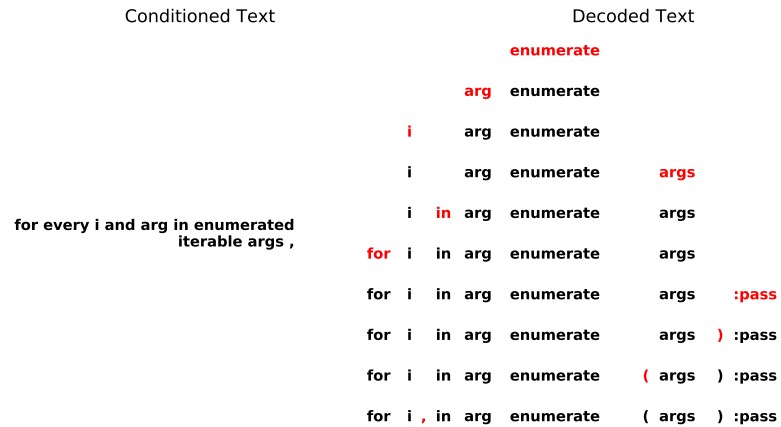

Figure 38: Generation order inferred by **Ours-Rare** for a pseudocode sample from the Django natural language to code test set with the sample id **431**.

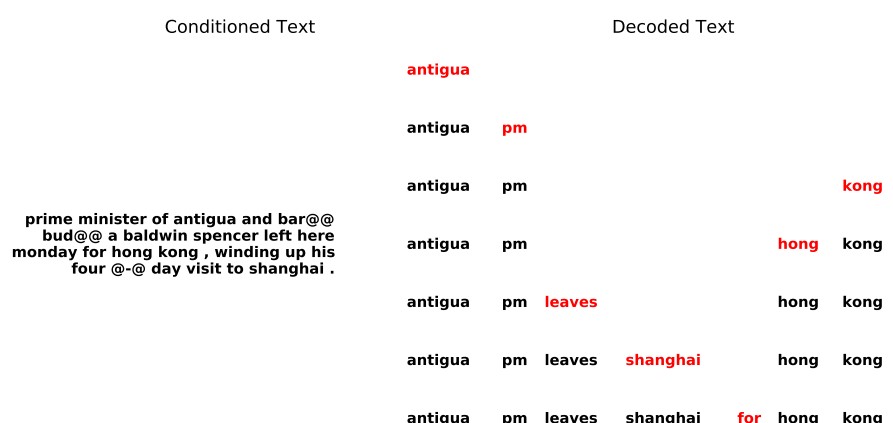

Figure 39: Generation order inferred by **Ours-VOI** for a text sample from the Gigaword text summarization test set with the sample id **15**.

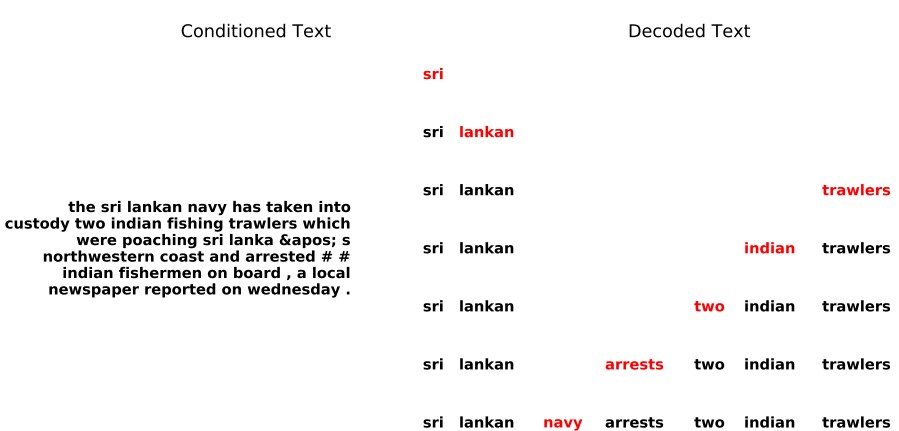

Figure 40: Generation order inferred by **Ours-VOI** for a text sample from the Gigaword text summarization test set with the sample id **33**.

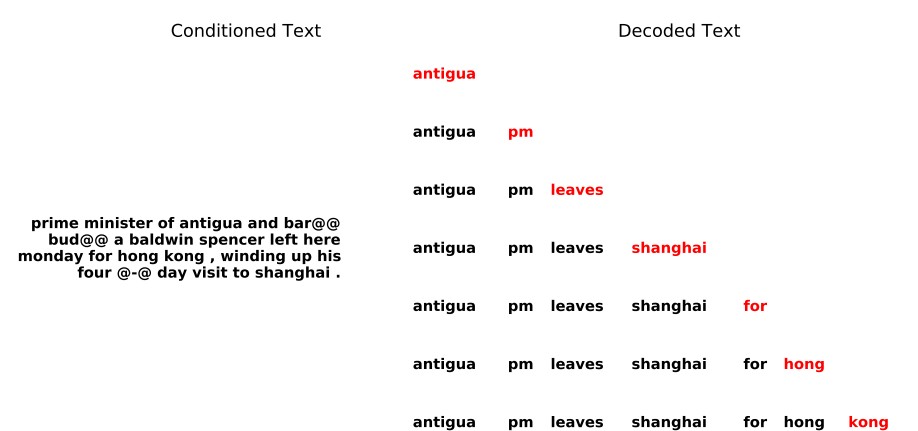

Figure 41: Generation order inferred by **Ours-L2R** for a text sample from the Gigaword text summarization test set with the sample id **15**.

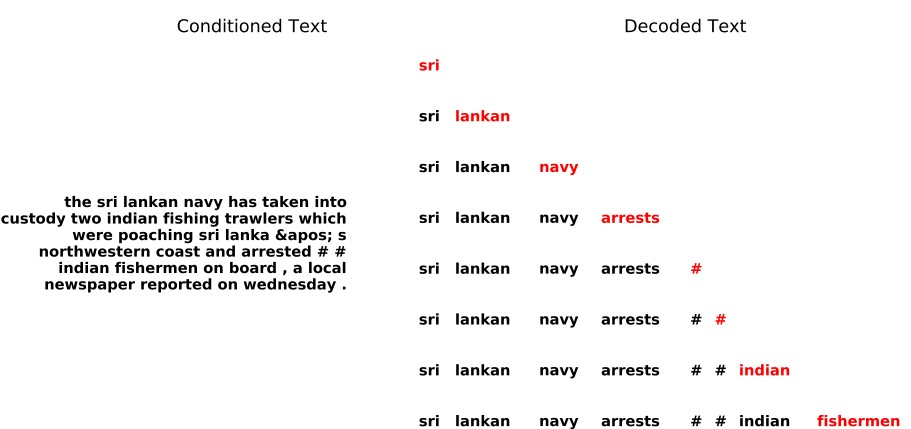

Figure 42: Generation order inferred by **Ours-L2R** for a text sample from the Gigaword text summarization test set with the sample id **33**.

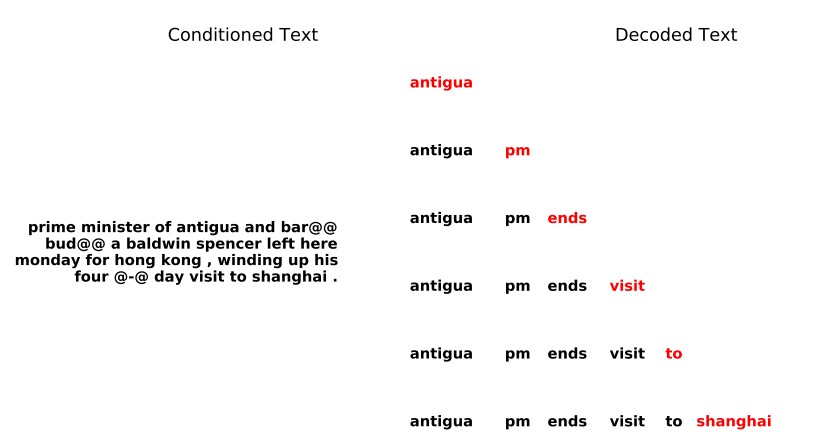

Figure 43: Generation order inferred by **Ours-Common** for a text sample from the Gigaword text summarization test set with the sample id **15**.

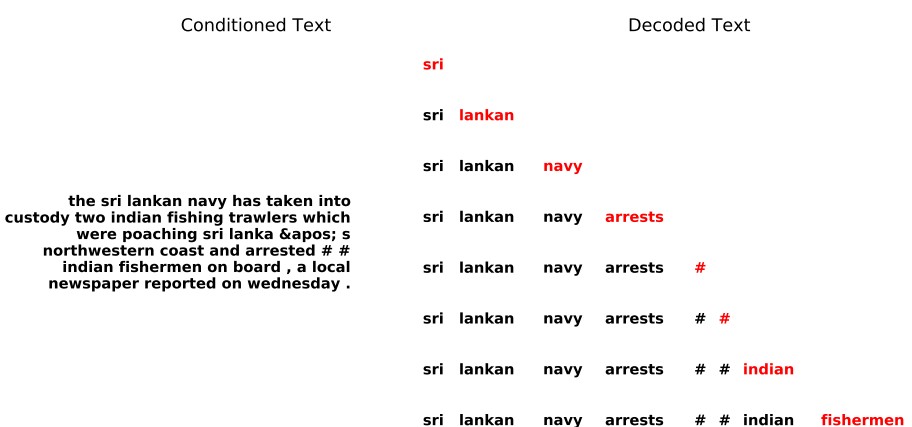

Figure 44: Generation order inferred by **Ours-Common** for a text sample from the Gigaword text summarization test set with the sample id **33**.

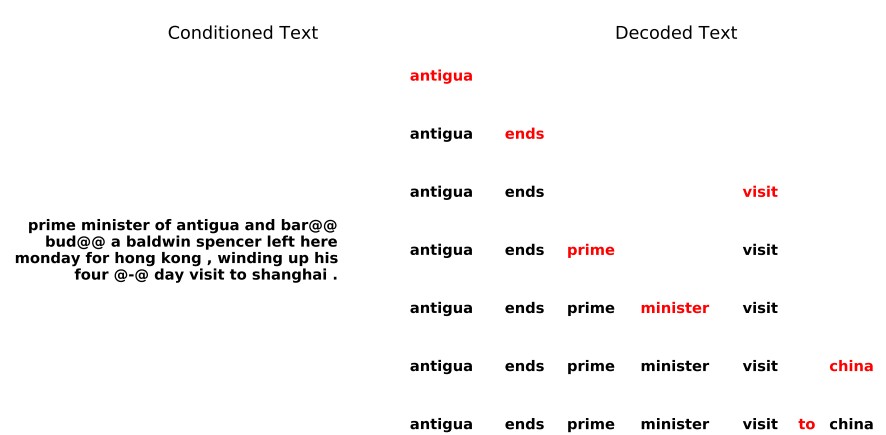

Figure 45: Generation order inferred by **Ours-Rare** for a text sample from the Gigaword text summarization test set with the sample id **15**.

Conditioned Text                                    Decoded Text

|  |  |  |  |  |  |  | **fishermen** |
|  |  |  |  | **arrests** |  |  | fishermen |
|  | **lankan** |  |  | arrests |  |  | fishermen |
the sri lankan navy has taken into custody two indian fishing trawlers which were poaching sri lanka ' s northwestern coast and arrested # # indian fishermen on board , a local newspaper reported on wednesday . |  | lankan | **navy** | arrests |  |  | fishermen |
|  | **sri** | lankan | navy | arrests |  |  | fishermen |
|  | sri | lankan | navy | arrests |  | **indian** | fishermen |
|  | sri | lankan | navy | arrests | **#** | indian | fishermen |
|  | sri | lankan | navy | arrests | # **#** | indian | fishermen |

Figure 46: Generation order inferred by **Ours-Rare** for a text sample from the Gigaword text summarization test set with the sample id **33**.

