# OpenReview forum: "Discovering Non-monotonic Autoregressive Orderings with Variational Inference"
_ICLR.cc/2021/Conference — ICLR 2021 Poster_

### Official Review · AnonReviewer1 · 2020-10-28

**Rating:** 6
**Confidence:** 4

**Review:**

This paper proposes to model the generation order as latent variables for sequence generation tasks, by optimizing the ELBO involving a proposed process of Variational Order Inference (VOI). To alleviate the difficulty of optimizing discrete latent variables, the authors propose to cast it as a one-step Markov Decision problem and optimize it using the policy gradient. The authors also introduce the recent developed Gumbel-matching techniques to derive the close-form of the posterior distribution.

Pros:
1. Overall, I think the research problem, i.e., explicit modeling the generation order,  in this work is interesting and worthy of discovering
2. Casting the optimization of discrete latent variables as a one-step MDP is interesting
3. Experiments show that the induced "best-first" order outperforms fixed orders, which verifies the motivation of the paper`
4. Extensive and inspiring analysis

Cons:
1. Explicit modeling the generation order is not a very novel idea that there have been many works on this topic.
2. For checking the generalization of the method and better comparison w/ InDIGO (though InDIGO also conducted on MSCOCO, Django and the current comparison is sufficiently fair), I would like to increase my rating if seeing more experiments on large scale machine translation benchmarks as those in InDIGO. This would also further support your claim of a general-purpose approach w/ little domain knowledge (if any).

-------
Minors:
- figure 1: could consider adding x, which would better match the descriptions of the paper (modeling p(y|x) instead of p(y))

Missing references:

[1] Chan, W., Kitaev, N., Guu, K., Stern, M. and Uszkoreit, J., 2019. KERMIT: Generative insertion-based modeling for sequences. arXiv preprint arXiv:1906.01604.

[2] Gu, J., Wang, C. and Zhao, J., 2019. Levenshtein transformer. In Advances in Neural Information Processing Systems (pp. 11181-11191).

[3] Bao, Y., Zhou, H., Feng, J., Wang, M., Huang, S., Chen, J. and Li, L., 2019. Non-autoregressive transformer by position learning. arXiv preprint arXiv:1911.10677.

---

> ### Author Response · Authors · 2020-11-23
> **Thank you for your review! Addressing concerns below [2/2]**
>
> >2. “For checking the generalization of the method and better comparison w/ InDIGO (though InDIGO also conducted on MSCOCO, Django and the current comparison is sufficiently fair), I would like to increase my rating if seeing more experiments on large scale machine translation benchmarks as those in InDIGO.”
>
> We would like to thank you for pointing us to this evaluation domain, and we have begun evaluating our method on the WMT Romanian-English machine translation task, which previous work has used for evaluation. It is our goal to include these translation results in our final submission, but given the limited amount of time and compute resource for rebuttals and the large number of training steps required, these translation results currently require time beyond the rebuttal period to be completed and entered into our submission. We will try to include these results in the final version of our paper.
>
> >3. Figure 1: could consider adding x, which would better match the descriptions of the paper (modeling p(y|x) instead of p(y))
>
> **In our updated version, we have updated Figure 1 to describe VOI as implemented for conditional sequence modeling to align with your suggestion.** If you have additional formatting or clarification suggestions, we are glad to further update our figures, and provide additional comments on the implementation of VOI.
>
> >Missing References
>
> We have added them in our related works section, and in Section 5 when we introduce the encoder and decoder architectures for conditional sequence generation.
>
> ---
>
> We hope these could address your concern, and we thank you again for the helpful comments. We are glad to discuss further comments and suggestions.

---

> ### Author Response · Authors · 2020-11-23
> **Thank you for your review! Addressing concerns below [1/2]**
>
> We sincerely thank you for your constructive comments, which are helpful in improving our submission. We would like to address the comments and questions below, and we have updated our submission accordingly.
>
> >1. “Explicit modeling the generation order is not a very novel idea that there have been many works on this topic.”
>
> We agree that modeling the generation order is not a novel concept by itself, and many works before have studied how to model the generation order. Indeed, our VOI's decoder network relies on a combination of Pointer Network + Transformer that several prior methods have utilized [1,2,3]. We are grateful to these works that came before us, because they have solved practical architectural issues that make the Pointer Network + Transformer models stable to train. Our use of Transformer-InDIGO as the decoder network demonstrates how solving practical architectural and stability issues can encourage future research (our research) in non-monotonic autoregressive modeling.
>
> While modeling generation order is not a novel concept by itself, this is only one of our contributions. Much like how prior work has made the Pointer Network + Transformer model architecture practical to train, we aim to streamline the inference of generation orders. We intend to replace inductive biases---such as model-pretraining [3], specialized loss functions [2], and generation order priors [1]---with an end-to-end data-driven method. **We believe that our contribution to creating a practical method that can infer high-quality generation orders without these inductive biases is novel.**
>
> **Another reason VOI is novel is because it is efficiently parallelized.** VOI's encoder network outputs generation orders in a single forward pass. An empirical comparison is presented in Section 5, Figure 2 of our updated paper, where we compare the runtime performance of VOI ($K=4$) with Searched Adaptive Order (SAO) on a single GPU in order to accurately measure the number of ops. Our speedup over SAO's inference time linearly increases with sequence length, and is 30 times faster than SAO for sequences of length 30. In our implementation, the time per training iteration for VOI is 6 times faster than SAO. In practice, as we distribute VOI across more GPUs, the penalty caused by the additional $K$ factor in our runtime is negligible.
>
> Thank you for making this point, as it has helped us elaborate on the positioning of our method within the larger space of generation order modeling. If you have additional suggestions, we'd love to continue this discussion with you within the rebuttal period.
>
> [1] Chan et al., KERMIT: Generative Insertion-Based Modeling for Sequences
>
> [2] Stern et al., Insertion Transformer: Flexible Sequence Generation via Insertion Operations
>
> [3] Gu et al., Insertion-based Decoding with automatically Inferred Generation Order

---

### Official Review · AnonReviewer3 · 2020-10-29

**Rating:** 7
**Confidence:** 2

**Review:**

The authors  propose the first domain-independent unsupervised learner that discovers high-quality autoregressive orders through fully-parallelizable end-to-end training without domain specific tuning.  Inspired by the variational auto-encoder, they propose an
encoder architecture to infer autoregressive orders. To solve  the non-differentiable ELBO ( discrete latent variables), they further construct a  practical algorithm with the help of policy gradients. The experiment results, such as the  global and local statistics for learned orders, are convincing.

Some problems.
1: The authors should clearly define the two similarity metrics between autoregressive orders in the appendix.
2: Please check up the X_axis in the Fig 3.
3: It would be better if the authors provide more results in the appendix,  such as the ablation studies about the 'K' and visualizations of sequences generated by the baselines.

---

> ### Author Response · Authors · 2020-11-23
> **Thank you for your review! Addressing concerns below**
>
> We sincerely thank you for your constructive comments, which are helpful in improving our submission. We would like to address the comments and questions below, and we have updated our submission accordingly.
>
> >1. Clearly Define The Two Similarity Metrics
>
> **In the updated version, we have included additional descriptions about the two similarity metrics in the Section 6 of the paper**. If you have additional suggestions to further improve our clarity, we are glad to discuss with you via OpenReview, and incorporate these suggestions into our paper.
>
> >2. X_axis in the Fig 3
>
> We thank you for recommending this clarifying detail. **In the updated version, we have now included, in Appendix E, a mapping from the X values in this Figure (it is now figure 4) to the full names of the parts of speech that these X values represent, along with example words.**
>
> >3. Ablation Studies About the 'K' and Visualizations
>
> **In Section 7 of our updated version, we have added instructions for choosing K, as well as an ablation study of the sensitivity of the encoder network to the value of K.** The main result of this study is that, while larger K generally produces better-performing encoder networks that fit more accurately to a ground truth order, K=4 appears to be sufficient, such that the encoder network’s performance is not significantly different from much larger K, while running fast.
>
> **We have also updated our submission with additional visualizations of sequences generated by our model, per your request, in Appendix F.** If you would like to see additional visualizations included, we are glad to generate more, and to iterate on the type of visualizations currently included in the paper.
>
> ---
>
> We hope these could address your concern, and we thank you again for the helpful comments. We are glad to discuss further comments and suggestions.

---

### Official Review · AnonReviewer2 · 2020-10-29
**Interesting problem,  proposed methods, and experimental results.**

**Rating:** 7
**Confidence:** 5

**Review:**

- Summary
This paper aims to decode both content and ordering of language models and proposes Variational Order Inference (VOI). The authors introduce a latent sequence variable z = (z_1, .. ,z_n) in which z_t is defined as the absolute position of the value generated. The authors model the posterior distribution of z as a Gumbel-Matching distribution which is relaxed as a Gumbel-Sinkorn distribution. To training the encoder and decoder networks, the ELBO is maximized using the policy gradient with baseline. The experimental results on Django and MS-COCO 2017 dataset show the proposed VOI outperforms the Transformer-InDIGO, as well as suggests that learned orders depend on content and best-first generation order.


- Strong points
	1. The research on non-autoregressive orders to generate language is interesting, and the proposed method using Gumbel-Sinkorn distribution is mathematically well sound and novel.
	2. The proposed method outperforms the previous Transformer-InDIGO and other baselines (Random, L2R, Common, Rare). This paper analyzed the learned orders globally and locally, and conducted ablation studies.
	3. The paper is well-written and the authors also provide source codes for reproducibility.

- Weak points
	1. The results are not compared with other SOTA auto-regressive algorithms.

- Questions
	- How do you think about transferring knowledge from auto-regressively trained such as GPT-2 to such non-autoregressive models? Using the pertained model for the encoder and decoder would improve the results? Have you tried this strategy?

---

> ### Author Response · Authors · 2020-11-23
> **Thank you for your review! Addressing concerns below**
>
> We sincerely thank you for your constructive comments, which are helpful in improving our submission. We would like to address the comments and questions below.
>
> >1. The results are not compared with other SOTA auto-regressive algorithms.
>
> We would like to first emphasize that the main focus of our work is to present a novel approach of learning non-sequential / non-monotonic orderings, which can generalize across different sequence modeling tasks and not rely on domain assumptions. We aim to discover autoregressive orderings in an end-to-end and efficient manner. **In this paper, we focus on studying the learned orderings and the generalizability of our approach across different tasks, and focus less on beating the SOTA methods on a particular challenge.**
>
> **We would also like to clarify that the autoregressive decoder network used in our method is modular by design.** It can be any model that supports non-monotonic sequence generation, and is **not restricted to Transformer-InDIGO**. In addition, we used Transformer-InDIGO trained with fixed orderings as a baseline because it is efficient to train the model with predefined fixed orderings, and training it using left-to-right (L2R) ordering is almost equivalent to training a typical Transformer, which provide meaningful and strong baselines. We also compare our model with the searched adaptive order (SAO) because it comes from previous work for discovering non-monotonic orderings, and represents a good baseline.
>
> Prior SOTA methods have proposed domain-specific improvements such as large-scale pretraining on other datasets and task-specific model architectures, and we believe such improvements can also be incorporated into our autoregressive decoder network. **Indeed, coupling our method with these techniques is a promising idea; however, we would like to leave that for future work and focus on providing a generalizable method for inferring generation orders that is efficient, learns high-quality orders, and is modularly compatible.**
>
> >Question: How do you think about transferring knowledge from auto-regressively trained such as GPT-2 to such non-autoregressive models? Using the pertained model for the encoder and decoder would improve the results?
>
> We believe it's a very nice idea to transfer knowledge / distill from auto-regressively pretrained models like GPT-2 to our model, and it poses an interesting research challenge. Such knowledge transfer has the potential to further improve the performance of VOI. **In fact, we found that VOI performed well without it.** For example, in the updated Gigaword results in Table 1, we find that our method outperforms, without domain-specific pre training or tuning, Transformer baselines with various fixed orderings. Our VOI performance is also approximately on par with that of PEGASUS-BASE [1], a larger Transformer model (their d_model = 3072, while our d_model = 2048) pretrained on 4 datasets and finetuned on Gigaword. As per to these findings, we believe that exploring knowledge transfer and distillation is out-of-scope for our current work and we would like to leave it for future work.
>
> ---
>
> We hope these could address your concern, and we thank you again for the helpful comments. We are glad to discuss further comments and suggestions.
>
> [1] Zhang et al., PEGASUS: Pre-training with Extracted Gap-sentences for Abstractive Summarization
>
> [2] Gu et al., Non-Autoregressive Neural Machine Translation

---

### Official Review · AnonReviewer4 · 2020-10-29
**Interesting idea on modeling generation orders with latent variables**

**Rating:** 6
**Confidence:** 3

**Review:**

This paper designed a new generative model by capturing the auto-regressive order as latent variables for sequence generation task. Based on combinatorical optimization techniques, the authors derived an policy gradient algorithm to optimize the variational lower bound. Empirical results on image caption and code generation showed that this method is superior than both fixed-order generation and previous adaptive-order method transformer-InDIGO. The authors further analyzed the learned orders on global and local level on COCO2017 dataset, demonstrating that the arrangement tend to follows the best-first strategy.


Concerns:
1. effect of sample size K: In section 5 training part, the paper claimed "For our model trained with Variational Order Inference , we sample K = 4 latents for each
training sample.". The sample size K is used to approximate the gradient in variational order inference and it also affects the training efficiency i.e. $O(NKdl^2)$. It's not clear how the author choose the appropriate sample size K. Some analysis or experiemnt reults on the sensitivity of sample size K will help clarify this concern.


2. experiments on nmt & running time: The papers didn't report any results on machine translation, an important task on conditional sequence generation. Since previous work(e.g. transformer-InDIGO) demonstrated superior results on several translation datasets, it's recommended that the authors also showed results on these datasets. Also one strength of the approach is its potential of fully parallelizing. A running time comparison will provide more convincing evidence to this claim.


3. figure: Figure 1. in section 4 showed the structure of variational order inference. Considering the paper is mainly focused on conditional generation, an conditional generation version will be better by incorporating x sequence into the figure.

---

> ### Author Response · Authors · 2020-11-23
> **Thank you for your review! Addressing concerns below**
>
> We sincerely thank you for your constructive comments, which are helpful in improving our submission. We would like to address the comments and questions below, and we have updated our submission accordingly.
>
> >1. Effect of Sample Size K
>
> **In Section 7 of our updated version, we have added instructions for choosing K, as well as an ablation study of the sensitivity of the encoder network to the value of K.** The main result of this study is that, while larger K generally produces better-performing encoder networks that fit more accurately to a ground truth order, K=4 appears to be sufficient, such that the encoder network’s performance is not significantly different from much larger K, while running fast.
>
> >2.1 Experiments on NMT
>
> We would like to thank you for pointing us to this evaluation domain, and we have begun evaluating our method on the WMT Romanian-English machine translation task. It is our goal to include these translation results in our final submission, but given the limited amount of time and compute resource for rebuttals and the large number of training steps required, these translation results currently require time beyond the rebuttal period to be completed and entered into our submission. We will try to include these results in the final version of our paper.
>
> >2.2 Running Time
>
> **In our updated version, we have added visualizations on *Time Per Training Iteration* and the *Generation Order Search Time* for our method and prior work.** We compare the runtime performance of VOI ($K=4$) with Searched Adaptive Order (SAO) on a single Tesla P100 GPU in order to accurately measure the number of ops required. We present these figures in Section 5. Since VOI outputs latent orderings in a single forward pass, we observe that VOI is significantly faster than SAO, which searches orderings sequentially. We found that VOI's speedup over SAO to infer a generation order linearly increases as the sequence length increases, and is 30 times faster than SAO for sequences of length 30. In our implementation, the training time per iteration for VOI is 6 times faster than SAO. The training penalty introduced by VOI while inferring generation orders is minimal. In practice, as we distribute VOI across more GPUs, the $K$ factor in the runtime is effectively divided by the number of GPUs used (if we ignore the parallelization overhead), so we can achieve further speedups.
>
> >3. Figure 1
>
> In our updated version, we have updated Figure 1 to explicitly describe the source sequence ‘x’.
>
> ---
>
> We hope these could address your concern, and we thank you again for the helpful comments. We are glad to discuss further comments and suggestions.

---

### Author Response · Authors · 2020-11-23
**Updates to our submission**

We thank all reviewers for their helpful comments and constructive feedback! Here we summarize the recent updates to our submission:

- **New results on Gigaword.** We have updated Table 1 with new results on Gigaword, which we finished after our initial paper submission. We report the Rouge-1, Rouge-2, and Rouge-L scores comparing our VOI with baselines. We find that, similar to MS-COCO and Django, our method outperforms baselines of different fixed orders.
- **Ablation on the choices of $K$.** In Section 7 of our updated version, we have added instructions for choosing $K$, the number of latent orderings to sample per data, as well as an ablation study of the sensitivity of the encoder network in VOI to the value of $K$.
- **Empirical runtime analysis of VOI.** In Section 5 of our updated version, we have added visualizations on *Time Per Training Iteration* and the *Generation Order Search Time*. We compare our VOI with Searched Adaptive Order (SAO) used for learning nonmonotonic orderings on Transformer-INDIGO. We implemented SAO according to the original paper's descriptions. We observe that, empirically, VOI can achieve significant speedup over SAO.
- **Updated VOI computation graph (Figure 1).** We have updated Figure 1 to make it clearer.
- **More visualizations.** We have added more visualizations, including those of VOI and fixed-order baselines, in Appendix F.
- **New analysis in Section 6.3** In our updated version, we now include an additional experiment in Section 6.3. We demonstrate that VOI learns autoregressive orders that not only depend on the target tokens $\mathbf y$, but also the content of the conditioning variable $\mathbf x$.

---

### Decision · Program_Chairs · 2021-01-07
**Final Decision**

**Decision:**

Accept (Poster)

**Comment:**

This paper deals with a particular model structure selection problem: inferring the order of a given sequence of latent variables. This problem is closely related to the matching problem that involves discrete optimization. The authors propose to cast the problem into a one-step Markov Decision problem and optimize it using the policy gradient.  The proposal here is using Variational Order Inference (VOI) using and using a Gumbel-Sinkhorn distribution to construct a proposal over approximate permutations. The approach is mathematically sound and novel.

Empirical results on image caption and code generation show promising results: method outperforms the previous Transformer-InDIGO and other baselines (Random, L2R, Common, Rare). This paper further analyzes the learned orders globally and locally, and conducts ablations.

The reviewers were overall very enthusiastic.